# CarbonSense: A Multimodal Dataset and Baseline for Carbon Flux Modelling

**Matthew Fortier**
Mila Quebec AI Institute
`matthew.fortier@mila.quebec`

**Mats L. Richter**
ServiceNow Research

**Oliver Sonnentag**
Département de géographie
Université de Montréal

**Chris Pal**
Mila Quebec AI Institute
Polytechnique Montréal
Canada CIFAR AI Chair

## Abstract

Terrestrial carbon fluxes provide vital information about our biosphere's health and its capacity to absorb anthropogenic $CO_2$ emissions. The importance of predicting carbon fluxes has led to the emerging field of data-driven carbon flux modelling (DDCFM), which uses statistical techniques to predict carbon fluxes from biophysical data. However, the field lacks a standardized dataset to promote comparisons between models. To address this gap, we present CarbonSense, the first machine learning-ready dataset for DDCFM. CarbonSense integrates measured carbon fluxes, meteorological predictors, and satellite imagery from 385 locations across the globe, offering comprehensive coverage and facilitating robust model training. Additionally, we provide a baseline model using a current state-of-the-art DDCFM approach and a novel transformer based model. Our experiments illustrate the potential gains that multimodal deep learning techniques can bring to this domain. By providing these resources, we aim to lower the barrier to entry for other deep learning researchers to develop new models and drive new advances in carbon flux modelling.

## 1 Introduction

The biosphere plays a critical role in regulating Earth's climate. Since the mid-20th century, terrestrial ecosystems have absorbed up to a third of anthropogenic carbon emissions [1]. However, climate change introduces uncertainty about the future resilience and capacity of these ecosystems. Understanding how the carbon dynamics of our biosphere are changing in response to both climate change and increasing anthropogenic pressures will give crucial insight into the health of our ecosystems and their ability to sequester carbon in the future.

Carbon fluxes describe the movement of carbon into and out of these ecosystems resulting from processes like photosynthesis and cellular respiration. They play a key role in assessing an ecosystem's health, but measuring them requires long-term field sensor deployment to cover an area of only 100-1000m$^2$ [2]. This bottleneck has given rise to the field of data-driven carbon flux modelling (DDCFM) where researchers use biophysical predictors such as meteorological and geospatial data to model carbon fluxes. By training on data from a variety of field sites in varying ecosystems, these models can be used to predict carbon fluxes regionally or globally [3, 4].

DDCFM presents a fascinating topic for deep learning researchers with real-world impact, yet it remains underexplored. As a result, the current state-of-the-art (SOTA) uses off-the-shelf solutions like random forests [5–7], gradient boosting [4], or ensembles of similar methods [8, 9]. These methods produce satisfactory results, but they fail to capitalize on the multimodal nature of the biophysical data. Recently, multimodal deep learning has exploded in popularity [10, 11] and may offer a more appropriate framework for DDCFM through effective data integration and advanced neural architectures. Such advances could significantly enhance the quality of information available to decision-makers, thereby improving our ability to address climate change.

We wish to lower the barriers to entry into this area and encourage more work on DDCFM from the deep learning community. Data preparation for DDCFM is currently performed ad-hoc by research teams, leading to inconsistency and lack of standardization. The absence of standardized datasets and benchmarks hinders reproducibility and comparability of research findings. Our work addresses these gaps with the following contributions:

- We provide an overview of DDCFM for deep learning researchers (Section 2)
- We publish a multimodal machine learning-ready (ML-ready) dataset for DDCFM with over 20 million hourly observations from 385 sites (Section 3)
- We provide a baseline model based on current SOTA practices in DDCFM, and compare it with a multimodal deep learning model which achieves improved performance (Section 4)

We will discuss our experiments in section 5 and provide guidelines for reporting results in this domain.

## 2 DATA-DRIVEN CARBON FLUX MODELLING

The application of machine learning techniques to model carbon fluxes was present as early as 2003 [12] where researchers used early neural networks for regional flux upscaling. Global upscaling models were first seen in 2011 [13], facilitated by the global research network FLUXNET which pooled carbon flux data from research teams around the world. Subsequent releases of FLUXNET increased the data quality and number of sites [14]. Recent models such as those developed by FLUXCOM [3][4] use a combination of meteorological data and geospatial data to improve model conditioning. Targeted modelling of specific regions such as subtropical wetlands [9] or arctic and boreal areas [7][8][5] allow researchers to develop a better understanding of the carbon cycle at these sites without concern for global generalization.

At its core, DDCFM is a regression problem. The target (carbon flux) depends on many factors including ecosystem composition, meteorological conditions, local topography and geology, and disturbances (fires, animal activity, etc). Meteorological data is relatively easy to obtain, but the other predictors are challenging to measure and represent, especially at a global scale. Remote sensing and semantic data are commonly employed as a proxy for the other predictors.

### 2.1 MEASURING FLUXES

The most common technique for measuring fluxes at ecosystem scale is eddy covariance (EC) [15]. This is a micrometeorological technique where researchers erect a tower (typically above canopy height) and mount sensors that measure atmospheric gas concentrations across small turbulent vortices (eddies). A simplified EC station is depicted in Figure 1. $CO_2$ and water vapour are the most widely measured, but some towers also measure methane ($CH_4$) [5, 6] or nitrous oxide ($N_2O$) [16]. Our work focuses on $CO_2$ due to the prevalance of standardized data collections.

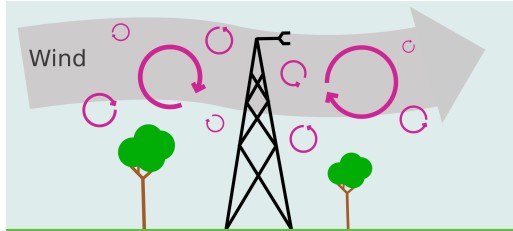

Figure 1: Simplified EC station. Sensors measure atmospheric gas concentrations across eddies.

Carbon fluxes are expressed as mass / area / time (ex $g \cdot m^{-2} \cdot hr^{-1}$). Gross primary productivity (GPP) refers to the total carbon uptake by plants through photosynthesis. Ecosystem respiration (RECO) is the total carbon returned to the atmosphere through both plant and microbial respiration. Net ecosystem exchange (NEE) is the small difference between the two large component fluxes, GPP and RECO; a carbon sink will have a negative NEE as more carbon is being consumed through GPP than released through RECO. NEE is the flux that is directly measured by EC stations and is the main focus of our experiments, but GPP and RECO (which are derived from NEE) are also provided in our dataset.

## 2.2 FLUX PREDICTORS

**Meteorological Data** DDCFM meteorological data comes from EC stations. In addition to carbon fluxes, EC stations measure local environmental and atmospheric conditions such as radiation, air temperature and relative humidity, precipitation, soil moisture and temperature, etc. The exact number and type of variables depends on the site, but regional networks maintain a minimum mandatory set for researchers wishing to submit their data [14]. For trained models looking to predict fluxes at the global scale, meteorological data can be obtained from publicly available reanalysis products such as ERA5 [17] which provides the variables on a 0.25-0.5 degree grid.

**Geospatial Data** Satellite imagery of the area surrounding an EC station can give useful information about the land cover and ecosystem makeup. The most common products for DDCFM are based on Moderate Resolution Imaging Spectroradiometer (MODIS) data [18]. This satellite pair ("Aqua" and "Terra") produce new imagery for Earth's surface every 1-2 days and have 36 spectral bands with resolutions varying between 250m and 1km. The MCD43A4 product is particularly common - it fuses MODIS data in a 16-day sliding window to produce a single image each day. This helps to address cloud coverage and produces images which remove angle effects from directional reflectance [19]. Each image therefore appears as it would from directly overhead at solar noon. MCD43A2 is also widely used, and contains categorical values for each pixel indicating snow and water cover [20]. The terms "geospatial data", "satellite data" and "remote sensing data" are often used interchangeably in this domain, but it should be noted that not all geospatial data comes from satellites.

**Semantic Data** Some models ingest semantic data such as land cover ("Croplands", "Evergreen needleleaf forest", "Snow and ice", etc). Land cover classifications follow standardized schemes such as the International Geosphere-Biosphere Programme (IGBP). Land cover classification is performed by domain experts, but some MODIS products coarsely approximate this information on a global grid [21], allowing this data to also be used for global inference.

## 3 THE CARBONSENSE DATASET

We present the first ML-ready dataset for DDCFM, CarbonSense. CarbonSense consists of EC station data and corresponding MODIS geospatial data for 385 sites across the globe, totalling over 27 million hourly observations. This section provides a brief overview of the dataset structure, processing pipeline, and usage guidelines. A more comprehensive guide is given in the supplementary material. For a detailed list of the 385 locations and their respective ecosystem types, see Appendix A.

### 3.1 DATA COLLECTION

All meteorological data was aggregated from major EC data networks, including FLUXNET 2015 [14], the Integrated Carbon Observation System (ICOS) 2023 release [22], ICOS Warm Winter release [23], and Ameriflux 2023 release [24]. These source datasets were chosen due to their use of the ONEFlux processing pipeline [14], ensuring standardized coding and units. A map of EC sites and their source networks is shown in Figure 2. North America and Europe are over-represented in this site list due greater data accessibility, and we discuss the implications of this in Section 3.3.

Geospatial data in CarbonSense are sourced from MODIS products. Specifically, we utilize the seven spectral bands from the MCD43A4 product [19], as well as the water and snow cover bands from MCD43A2 [20]. Following the guidelines from [18], we extract images in a 4km by 4km square centered on each EC station. Given a spatial resolution of 500m per pixel, this yields an 8x8 pixel image with 9 channels for every site-day.

### 3.2 DATA PIPELINE

The first stage in the pipeline is EC data fusion. Many sites had overlapping data from different source networks. For example, the site Degero in Sweden (SE-Deg) had data from 2001-2020 in the ICOS Warm Winter release, and data from 2019-2022 in the ICOS 2023 release. Data were fused with overlapping values taken from the more recent release as in previous DDCFM work [4]. Any

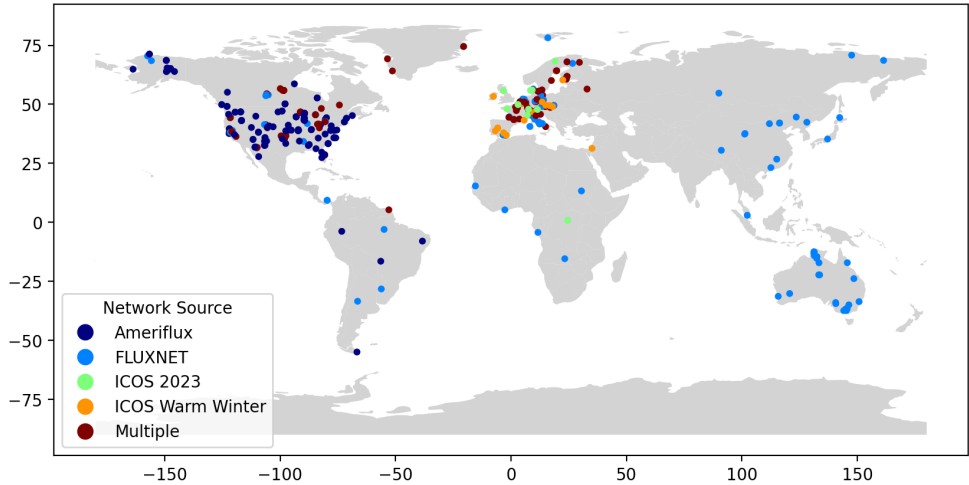

Figure 2: Global map of eddy covariance sites used in CarbonSense, with corresponding source networks. Some sites were present in multiple networks.

sites which report half-hourly data were downsampled to hourly at this stage, and daily and monthly recordings were discarded.

Once fused, we extracted the relevant time blocks for each EC station along with its geographic location. This metadata was used to obtain the appropriate MODIS data for each site. Data was pulled procedurally from Google Earth Engine [25].

Meteorological data was pruned to remove unwanted variables. Some, like soil moisture and temperature, were either unavailable for most sites or were heavily gap-filled. We removed these variables to reduce the risk of compounding errors on the underlying pipeline gapfilling techniques. A full list of variables at this stage is given in Table 6.

As a final stage in the pipeline, we apply a min-max normalization on predictor variables. We map cyclic variables (those with a cyclic range such as wind direction) to the range $[-1, 1)$ and acyclic variables to the range $[-0.5, 0.5)$. This normalization procedure is conducive to our Fourier encoding method discussed in Section 4.1.

We offer CarbonSense as a finished dataset but also provide the raw data. The full pipeline code is available so that researchers can run and modify it freely. Our pipeline can be configured to include other variables or to have different "leniency" for gap-filled values. For example, those who wish to use CarbonSense with strictly observed values may do so at the cost of a smaller number of samples. A diagram of the entire pipeline is shown in Figure 3.

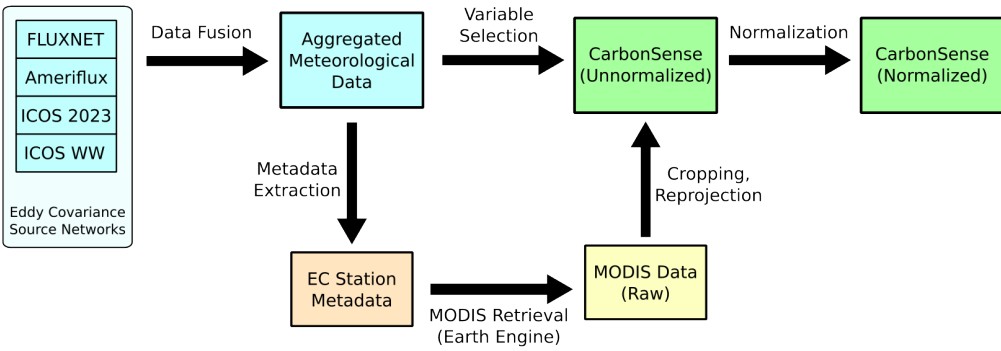

Figure 3: Data pipeline used to create CarbonSense from EC and MODIS data.

### 3.3 Using the Dataset

**Site Sampling** The biased geographic and ecological distribution of sites remains a challenge in DDCFM, and CarbonSense is no different. Given the significant overrepresentation of certain regions (North America, Europe) and ecosystems (evergreen needleleaf forests, grasslands), we maintain a partitioned structure where each site has its own directory containing EC data, geospatial data, and metadata. Researchers are encouraged to select sites for training and testing based on their experiment objectives such as high performance on particular ecosystems, or out-of-distribution generalization. Our experiments in section 5 are an example of the latter.

**Dataloader** We supply an example PyTorch dataloader for CarbonSense specifically tailored to our model. Using the dataloader requires specifying which carbon flux to use as the target (ex NEE, GPP, RECO), which sites to include in each dataloader instance, and the context window length for multi-timestep training.

**Licensing** CarbonSense is available under the CC-BY-4.0 license, meaning it can be shared, transformed, and used for any purpose given proper attribution. This is an extension of the same license for all three source networks, and MODIS data is provided under public domain. We feel that permissive licensing is essential in order to foster greater scientific interest in DDCFM.

## 4 The EcoPerceiver Architecture

In this section we present EcoPerceiver, a multimodal architecture for DDCFM. The SOTA for DDCFM are tabular methods, and we felt it would be appropriate to include a baseline model which demonstrates how deep learning concepts can be leveraged for this unique problem domain.

EcoPerceiver is based on the Perceiver architecture [26], which cross attends a variable number of inputs onto a compact latent space, allowing for extreme input flexibility. Missing inputs are common in DDCFM due to coverage gaps, outlier values, or failing sensors. Rather than rely on gapfilling techniques, we chose this architecture for its robustness to missing inputs.

We also wanted a model which could ingest data from a varying time window. To our knowledge, this is the first DDCFM model to treat carbon dynamics as non-Markovian with respect to predictors. We feel this more accurately reflects biological processes, since a plant's rate of photosynthesis may also depend on conditions hours or days into the past. Our ablation experiments in Appendix B.7 explore this idea further.

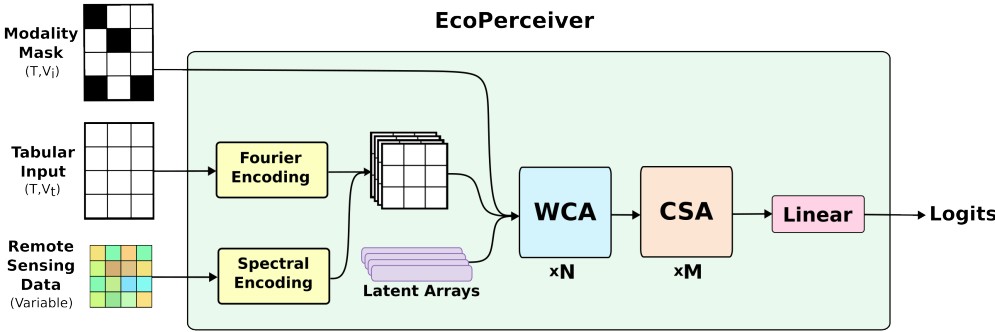

Figure 4: Overview of EcoPerceiver architecture.

### 4.1 Data Ingestion

Small fluctuations in meteorological variables have the potential to influence ecological processes. For this reason, it is important that the model is sensitive to small changes in input values. We take inspiration from NeRF's Fourier encoding [27] which maps continuous values to higher dimensional

space with high frequency sinusoids. Each variable $x$ is therefore encoded as:

$$f(x; K) = \left[ \ldots, \sin(2^k \pi x), \cos(2^k \pi x), \ldots \,\middle|\, k \in [0, K) \right], \tag{1}$$

where $K$ is a hyperparameter indicating the maximum sampling frequency. Higher values of $K$ allow the model to better discern between small differences in input. With our normalization scheme, cyclic variables at values of $-1$ and $1$ will produce identical vectors under this transform as intended. Each input is given a learned embedding specific to the underlying variable. This is then concatenated with the Fourier encoding to produce a final input vector of length $H_i = 2K + l_{emb}$ for each input. Figure 5 depicts this encoding procedure.

Geospatial data is similarly processed, except that each spectral band is flattened and mapped into a vector of length $2K$ via linear transformation instead of Fourier encoding. Each band is then concatenated with an embedding to produce a vector of length $H_i$. We then stack the encoded data to create a matrix of shape $(T, V_t, H_i)$ where $V_t$ is the total number of variables (tabular values + spectral bands) and $T$ is the context window length.

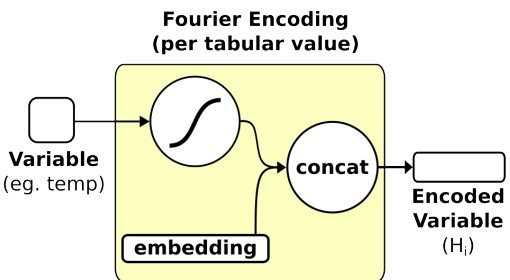

To account for missing values and timesteps without geospatial data, EcoPerceiver takes a modality mask indicating which values to ignore in the cross attentive layers. This modality mask doubles as a dropout mechanism which reduces over-reliance on a small subset of variables (observational dropout).

Figure 5: Fourier input encoding for EcoPerceiver. Spectral inputs are similarly processed, but with a linear projection instead of Fourier encoding.

## 4.2 WINDOWED CROSS ATTENTION

We build on Perceiver's core concept of cross-attending data onto a compact latent space for processing. EcoPerceiver uses a latent space of size $(T, H_l)$ where $H_l$ is the latent hidden dimension. Each token extracts input data via cross-attention from its respective timestep's observations. Intuitively, each token may represent the ecosystem's "state" at a particular time, and ingests observations from that timestep.

This operation would be inefficient with vanilla cross attention, as each token would use at most $\frac{1}{T}$ observations with an attention mask removing the rest. We take inspiration from SWin Transformer [28] and instead push the context window dimension ($T$) into the batch dimension for both input and latent space. The resulting Windowed Cross Attention (WCA) has a runtime of $O(T \cdot V_t \cdot H_a)$ where $H_a$ is the projection dimension.

In keeping with Perceiver, each WCA operation is followed by a self-attention operation in the latent space. We pass a causal mask to the self attention so each timestep is conditioned only on past and present observations. We refer to this as Causal Self Attention (CSA). This constitutes a full WCA block as shown in Figure 6.

WCA blocks are repeated $N$ times, repeatedly cross attending inputs onto the latent space with self attention in between. We then apply a series of $M$ CSA operations and use the final timestep's token as input to a linear layer. The output of this is the estimate of the desired carbon flux.

## 5 EXPERIMENTS

In this section, we present a series of experiments using CarbonSense. Our primary analysis includes two models: an EcoPerceiver model as introduced in 4, and an XGBoost model [29] implemented to mimic current SOTA approaches in DDCFM [4]. We demonstrate the power of tailored deep architectures for DDCFM and establish a robust baseline that will support and inspire future research efforts. We also present guidelines for running similar experiments and presenting results. Further model comparisons are explored in Appendix B.6 alongside ablation studies and qualitative analysis.

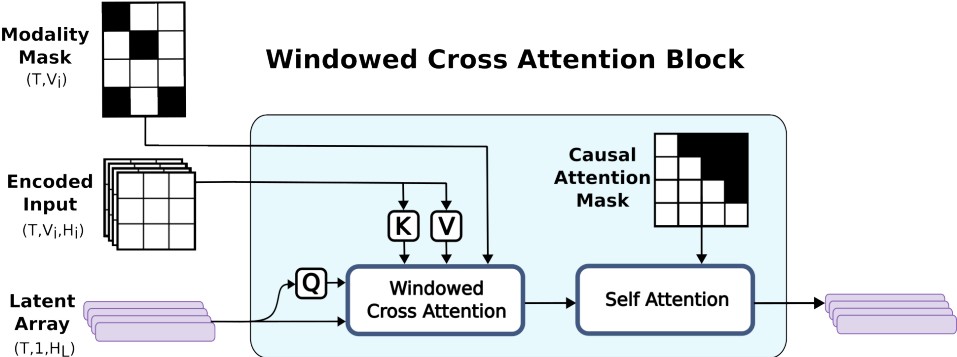

Figure 6: Windowed Cross Attention (WCA) block. Encoded inputs are cross-attended onto the latent space with a modality mask to indicate missing values. The time dimension is pushed into the batch dimension, so this operation is performed $B \cdot T$ times per batch. Causal self attention proceeds as normal.

## 5.1 Data Splitting

EC stations were randomly divided into train and test sets based on their IGBP ecosystem classification (IGBP type). We mostly refer to IGBP types by their acronyms for brevity, but a list of IGBP types with expanded names is found in Appendix A.

Despite the imbalance of IGBP types in CarbonSense, we wanted the test set to be as balanced as possible. The number of sites in the test set were determined with $\min(5, \lceil 0.2 * \text{num\_sites} \rceil)$. This provided between 1 and 5 sites per IGBP type as shown in Table 1. As a consequence, SNO and DNF provide information about zero-shot generalization, and CVM and WAT about one-shot generalization.

The main focus of this research is on models trained *across* different ecosystem types, as opposed to other research studying DDCFM *within a single* type (ex [5, 8, 9]). However, the partitioned nature of CarbonSense makes it flexible for different modelling objectives, such as individual ecosystems. We give an example of this in Appendix B.8.

Table 1: Train / test split distribution by IGBP type

| IGBP | Train | Test |
|------|-------|------|
| WET | 42 | 5 |
| DNF | 0 | 1 |
| WSA | 8 | 2 |
| EBF | 10 | 3 |
| ENF | 80 | 5 |
| DBF | 42 | 5 |
| CRO | 44 | 5 |
| MF | 10 | 3 |
| GRA | 59 | 5 |
| OSH | 25 | 5 |
| CVM | 1 | 1 |
| CSH | 5 | 2 |
| SAV | 11 | 3 |
| SNO | 0 | 1 |
| WAT | 1 | 1 |

## 5.2 Model Configurations

EcoPerceiver experiments were each run on 4 A100 GPUs using dataset parallelization. The train sites were further divided into train and validation splits at a 0.8 / 0.2 ratio respectively. We used the AdamW optimizer [30] with a learning rate of 8e-5 and a batch size of 4096. A single warm-up epoch was performed followed by a cosine annealing learning rate schedule over 20 epochs, but all experiments converged between 6 and 13 epochs.

XGBoost experiments were run on CPU nodes. We designed our XGBoost experiments to resemble [4] as closely as possible. This allows us to compare EcoPerceiver's relative performance against a stand-in for the SOTA. Appendix B provides a detailed description of XGBoost data preprocessing, hyperparameters for all models, ablation studies, and more.

## 5.3 METRICS

The most commonly used performance metric in DDCFM (and any form of hydrologic modelling) is the Nash-Sutcliffe Modelling Efficiency (NSE) [31], described with the following equation:

$$\text{NSE}(x) = 1 - \frac{\sum_i (y_i - x_i)^2}{\sum_i (y_i - \bar{y})^2} \tag{2}$$

where a value of 1 represents perfect correlation between $x$ and $y$. A value of 0 represents the same performance as guessing the mean of $y$, and negative values indicate that the mean of $y$ is a better predictor than $x$. NSE is more challenging to use directly as a loss function since it would require the dataloader to also provide the mean of the data for a given site or ecosystem type. We therefore use mean squared error (MSE) as a loss function and report its root (RMSE) as well as NSE in our results. Data balance in results reporting is also a concern. At first glance, the data appears very imbalanced with respect to ecosystem prevalence. CarbonSense contains 64 grasslands (GRA) sites, but only 1 deciduous needleleaf forest (DNF). While this is an extreme gap, ecosystems are more diverse than IGBP types can capture; grasslands in central North America will differ significantly from those in Europe or Asia. Still, it is prudent to separate results by IGBP type to give a better picture of model performance.

## 5.4 RESULTS

We ran 10 experiments with each model using different seeds to get an accurate picture of performance. Table 2 shows the mean performance on the test set for each model over every IGBP type, and we give an example of model output visualization in Figure 7.

Table 2: NSE and RMSE by model and IGBP type, aggregate mean across 10 seeds. Bold numbers indicate better performance. Pairwise t-test results are given for NSE values with 9 degrees of freedom.

| | XGBoost | | EcoPerceiver | | t-test (NSE) | |
|---|---|---|---|---|---|---|
| IGBP | NSE | RMSE | NSE | RMSE | t-statistic | p-value |
| CRO | 0.8066 | 3.2381 | **0.8482** | **2.8677** | 13.4689 | 0.0000 |
| CSH | 0.7510 | 1.5224 | **0.7670** | **1.4709** | 1.9947 | 0.0772 |
| CVM | 0.5277 | 5.5157 | **0.5763** | **5.2236** | 9.6586 | 0.0000 |
| DBF | 0.7250 | 4.0959 | **0.7547** | **3.8678** | 10.5993 | 0.0000 |
| DNF | 0.2803 | 4.0974 | **0.4336** | **3.6322** | 8.6338 | 0.0000 |
| EBF | 0.7966 | 4.6050 | **0.8220** | **4.3070** | 8.1990 | 0.0000 |
| ENF | **0.7765** | **2.8141** | 0.7694 | 2.8579 | -2.3853 | 0.0409 |
| GRA | 0.7461 | 3.2487 | **0.7967** | **2.9059** | 13.7609 | 0.0000 |
| MF | 0.7559 | 3.8633 | **0.7717** | **3.7361** | 8.3540 | 0.0000 |
| OSH | 0.5451 | 1.8796 | **0.6060** | **1.7475** | 3.9356 | 0.0034 |
| SAV | 0.5802 | 1.6514 | **0.7368** | **1.3070** | 28.0814 | 0.0000 |
| SNO | -0.0370 | 1.4291 | **0.2898** | **1.1816** | 16.3974 | 0.0000 |
| WAT | **-11.0524** | **3.1838** | -14.4010 | 3.5802 | -2.4809 | 0.0349 |
| WET | **0.4530** | **2.2073** | 0.4137 | 2.2830 | -2.1005 | 0.0651 |
| WSA | 0.6132 | 2.5153 | **0.6267** | **2.4706** | 2.6798 | 0.0252 |

EcoPerceiver outperformed the XGBoost baseline across most IGBP types. XGBoost performed better in permanent wetlands (WET), water bodies (WAT), and evergreen needleleaf forests (ENF) by a slim margin. These are the three key ecosystems of the boreal biome, indicating XGBoost retains an advantage in that region. WAT is particularly far out of distribution (EC stations are mounted above lakes) and both models did worse than predicting the mean, indicating this could be an issue with data quantity.

Besides WAT, EcoPerceiver did substantially better on zero- and one-shot tests. High predictive power on out-of-distribution sites like this is especially important for researchers wishing to run inference on global data, where each grid cell is likely to be quite different from any of the training sites.

Our results also underline the importance of using NSE as the main metric for evaluation. Consider the models' performance on open savannas (SAV). XGBoost had an RMSE of 1.6514 versus EcoPerceiver's 1.3070. The magnitude of difference is small, and both values are significantly lower than the RMSE of many other IGBP types. But XGBoost had an NSE of 0.5802 while EcoPerceiver achieved 0.7368 which is a significant improvement. Different ecosystems have different variances in their carbon fluxes, and NSE accounts for this by dividing the performance by the variance of the target.

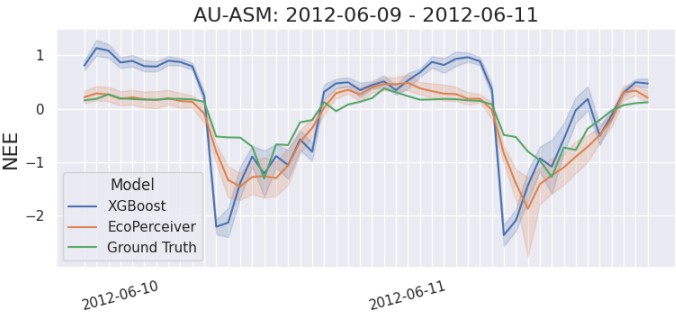

Figure 7: Example NEE measurements and model predictions from an Australian cropland site. Qualitative analysis of model outputs is important in DDCFM, and is discussed in Appendix B.9.

## 6  CONCLUSION

Our work establishes a foothold for deep learning in the field of DDCFM. We provide an open source ML-ready dataset, CarbonSense, using EC station data and geospatial data from a variety of ecosystems. DDCFM is inherently a multimodal task, and our baseline model EcoPerceiver demonstrates that recent advances in multimodal deep learning can unlock substantial performance gains in this domain. We implore more deep learning researchers to help develop this field further, because the potential of artificial intelligence to improve our world can only be realized if we actively apply it to solve pressing social and environmental issues.

**Future Work**   Our work leaves much to be explored in both dataset and model development. CarbonSense can be expanded as more EC station data is incorporated into regional network releases. Additional geospatial data could be added in the form of global soil products or higher resolution satellite imagery. Future models may work to address the shortcomings of EcoPerceiver in the boreal ecosystems, or incorporate more sophisticated fusion techniques like the use of convolutional layers in image ingestion.

**Limitations**   Data diversity remains the biggest challenge in this domain. CarbonSense has a data imbalance in not only ecosystem types, but geographic location. Africa, Central Asia, and South America are all underrepresented. While these areas contain many EC stations, most do not have readily available data in ONEFlux format, presenting a barrier to their inclusion in CarbonSense. Researchers should be aware of the consequences of developing models with imbalanced data, including poor performance in underrepresented areas.

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

# Appendices

## A  EDDY COVARIANCE SITE DETAILS

Here we provide an exhaustive list of EC sites used in CarbonSense along with their most recent publication. As per Ameriflux's data policy, each site has an individual citation with DOI; other networks simply required citation of the unified release. It would be impractical to have each site's full description in these tables, but the first two letters of each code represent the country where the site is located (ex. "DE" for Germany).

We also enumerate all meteorological predictors and targets in Table 6, and provide a temporal distribution of EC data by year in Figure 8.

Table 3: EC Sites

**Croplands (CRO)**

| | | | | | |
|---|---|---|---|---|---|
| BE-Lon [22] | CA-ER1 [32] | CA-MA1 [33] | CA-MA2 [34] | CH-Oe2 [23] | CZ-KrP [23] |
| DE-Geb [22] | DE-Kli [22] | DE-RuS [22] | DE-Seh [14] | DK-Fou [14] | DK-Vng [22] |
| FI-Jok [14] | FI-Qvd [23] | FR-Aur [22] | FR-EM2 [22] | FR-Gri [22] | FR-Lam [22] |
| IT-BCi [23] | IT-CA2 [14] | US-A74 [35] | US-ARM [36] | US-Bi1 [37] | US-Bi2 [38] |
| US-CF1 [39] | US-CF2 [40] | US-CF3 [41] | US-CF4 [42] | US-CRT [43] | US-CS1 [44] |
| US-CS3 [45] | US-CS4 [46] | US-DFC [47] | US-DS3 [48] | US-Lin [49] | US-Mo1 [50] |
| US-Mo3 [51] | US-Ne1 [52] | US-RGA [53] | US-RGB [54] | US-RGo [55] | US-Ro1 [56] |
| US-Ro2 [57] | US-Ro5 [58] | US-Ro6 [59] | US-Tw2 [60] | US-Tw3 [61] | US-Twt [14] |
| US-xSL [62] | | | | | |

**Closed Shrublands (CSH)**

| | | | | | |
|---|---|---|---|---|---|
| BE-Maa [22] | IT-Noe [14] | US-KS2 [63] | US-Rls [64] | US-Rms [65] | US-Rwe [66] |
| US-Rwf [67] | | | | | |

**Cropland/Natural Vegetation Mosaics (CVM)**

| | |
|---|---|
| US-HWB [68] | US-xDS [69] |

**Deciduous Broadleaf Forests (DBF)**

| | | | | | |
|---|---|---|---|---|---|
| AU-Lox [14] | BE-Lcr [22] | CA-Cbo [70] | CA-Oas [14] | CA-TPD [71] | CZ-Lnz [22] |
| CZ-Stn [23] | DE-Hai [22] | DE-Hzd [23] | DE-Lnf [14] | DK-Sor [22] | FR-Fon [22] |
| FR-Hes [22] | IT-BFt [22] | IT-CA1 [14] | IT-CA3 [14] | IT-Col [14] | IT-Isp [14] |
| IT-PT1 [14] | IT-Ro1 [14] | IT-Ro2 [14] | JP-MBF [14] | MX-Tes [72] | PA-SPn [14] |
| US-Bar [73] | US-Ha1 [74] | US-MMS [75] | US-MOz [76] | US-Oho [77] | US-Rpf [78] |
| US-UMB [79] | US-UMd [80] | US-WCr [14] | US-Wi1 [81] | US-Wi3 [82] | US-Wi8 [83] |
| US-xBL [84] | US-xBR [85] | US-xGR [86] | US-xHA [87] | US-xML [88] | US-xSC [89] |
| US-xSE [90] | US-xST [91] | US-xTR [92] | US-xUK [93] | ZM-Mon [14] | |

**Deciduous Needleleaf Forests (DNF)**

| |
|---|
| BR-CST [94] |

**Evergreen Broadleaf Forests (EBF)**

| | | | | | |
|---|---|---|---|---|---|
| AU-Cum [14] | AU-Rob [14] | AU-Wac [14] | AU-Whr [14] | AU-Wom [14] | BR-Sa3 [14] |
| CN-Din [14] | FR-Pue [22] | GF-Guy [22] | GH-Ank [14] | IT-Cp2 [22] | IT-Cpz [14] |
| MY-PSO [14] | | | | | |

Table 4: EC Sites (cont'd)

**Evergreen Needleleaf Forests (ENF)**

| | | | | | |
|---|---|---|---|---|---|
| AR-Vir [14] | CA-Ca1 [95] | CA-Ca2 [96] | CA-LP1 [97] | CA-Man [14] | CA-NS1 [98] |
| CA-NS2 [99] | CA-NS3 [100] | CA-NS4 [101] | CA-NS5 [102] | CA-Obs [14] | CA-Qfo [103] |
| CA-SF1 [104] | CA-SF2 [105] | CA-TP1 [106] | CA-TP2 [14] | CA-TP3 [107] | CA-TP4 [14] |
| CH-Dav [22] | CN-Qia [14] | CZ-BK1 [22] | CZ-RAJ [23] | DE-Lkb [14] | DE-Msr [22] |
| DE-Obe [23] | DE-RuW [22] | DE-Tha [22] | DK-Gds [22] | FI-Hyy [22] | FI-Ken [22] |
| FI-Let [22] | FI-Sod [14] | FI-Var [22] | FR-Bil [22] | FR-FBn [23] | FR-LBr [14] |
| IL-Yat [23] | IT-La2 [14] | IT-Lav [23] | IT-Ren [22] | IT-SR2 [22] | IT-SRo [14] |
| NL-Loo [14] | RU-Fy2 [23] | RU-Fyo [23] | SE-Htm [22] | SE-Nor [22] | SE-Ros [23] |
| SE-Svb [22] | US-BZS [108] | US-Blo [14] | US-CS2 [109] | US-Fmf [110] | US-Fuf [111] |
| US-GBT [14] | US-GLE [112] | US-HB2 [113] | US-HB3 [114] | US-Ho2 [115] | US-KS1 [116] |
| US-Me1 [117] | US-Me2 [118] | US-Me3 [119] | US-Me4 [14] | US-Me5 [14] | US-Me6 [120] |
| US-NC1 [121] | US-NC3 [122] | US-NR1 [123] | US-Prr [14] | US-Vcm [124] | US-Vcp [125] |
| US-Wi0 [126] | US-Wi2 [14] | US-Wi4 [127] | US-Wi5 [128] | US-Wi9 [129] | US-xAB [130] |
| US-xBN [131] | US-xDJ [132] | US-xJE [133] | US-xRM [134] | US-xSB [135] | US-xTA [136] |
| US-xYE [137] | | | | | |

**Grasslands (GRA)**

| | | | | | |
|---|---|---|---|---|---|
| AT-Neu [14] | AU-DaP [14] | AU-Emr [14] | AU-Rig [14] | AU-Stp [14] | AU-TTE [14] |
| AU-Ync [14] | BE-Dor [23] | CA-MA3 [138] | CH-Aws [23] | CH-Cha [23] | CH-Fru [23] |
| CH-Oe1 [14] | CN-Cng [14] | CN-Dan [14] | CN-Du2 [14] | CN-Du3 [14] | CN-HaM [14] |
| CN-Sw2 [14] | CZ-BK2 [14] | DE-Gri [22] | DE-RuR [22] | DK-Eng [14] | FR-Mej [22] |
| FR-Tou [22] | GL-ZaH [22] | IT-MBo [23] | IT-Niv [22] | IT-Tor [22] | NL-Hor [14] |
| PA-SPs [14] | RU-Ha1 [14] | SE-Deg [22] | US-A32 [139] | US-AR1 [140] | US-AR2 [141] |
| US-ARb [142] | US-ARc [143] | US-BRG [144] | US-Cop [145] | US-Goo [14] | US-Hn2 [146] |
| US-IB2 [14] | US-KFS [147] | US-KLS [148] | US-Kon [149] | US-Mo2 [150] | US-NGC [151] |
| US-ONA [152] | US-Ro4 [153] | US-SRG [154] | US-Seg [155] | US-Sne [156] | US-Snf [157] |
| US-Var [158] | US-Wkg [159] | US-xAE [160] | US-xCL [161] | US-xCP [162] | US-xDC [163] |
| US-xKA [164] | US-xKZ [165] | US-xNG [166] | US-xWD [167] | | |

**Mixed Forests (MF)**

| | | | | | |
|---|---|---|---|---|---|
| AR-SLu [14] | BE-Bra [22] | BE-Vie [22] | CA-Gro [168] | CD-Ygb [22] | CH-Lae [23] |
| CN-Cha [14] | DE-Har [22] | DE-HoH [22] | JP-SMF [14] | US-Syv [169] | US-xDL [170] |
| US-xUN [171] | | | | | |

**Open Shrublands (OSH)**

| | | | | | |
|---|---|---|---|---|---|
| CA-NS6 [172] | CA-NS7 [24] | CA-SF3 [14] | ES-Agu [23] | ES-Amo [14] | ES-LJu [23] |
| ES-LgS [14] | ES-Ln2 [14] | GL-Dsk [22] | IT-Lsn [22] | RU-Cok [14] | US-EML [173] |
| US-Fcr [174] | US-Hn3 [175] | US-ICh [176] | US-ICt [177] | US-Jo1 [178] | US-Jo2 [179] |
| US-Rws [180] | US-SRC [181] | US-Ses [182] | US-Sta [14] | US-Whs [183] | US-Wi6 [184] |
| US-Wi7 [185] | US-xHE [186] | US-xJR [187] | US-xMB [188] | US-xNQ [189] | US-xSR [190] |

**Savannas (SAV)**

| | | | | | |
|---|---|---|---|---|---|
| AU-ASM [14] | AU-Cpr [14] | AU-DaS [14] | AU-Dry [14] | AU-GWW [14] | CG-Tch [14] |
| ES-Abr [23] | ES-LM1 [23] | ES-LM2 [23] | SD-Dem [14] | SN-Dhr [14] | US-LS2 [191] |
| US-Wjs [192] | US-xSJ [193] | | | | |

**Snow and Ice (SNO)**

| | | | | | |
|---|---|---|---|---|---|
| US-NGB [194] | | | | | |

Table 5: EC Sites (cont'd)

**Water Bodies (WAT)**

| US-Pnp [195] | US-UM3 [196] | | | | |
|---|---|---|---|---|---|

**Permanent Wetlands (WET)**

| AR-TF1 [197] | AU-Fog [14] | CA-ARB [198] | CA-ARF [199] | CA-CF1 [200] | CA-DB2 [201] |
|---|---|---|---|---|---|
| CA-DBB [202] | CN-Ha2 [14] | CZ-wet [22] | DE-Akm [23] | DE-SfN [14] | DE-Spw [14] |
| DE-Zrk [14] | DK-Skj [22] | FI-Lom [14] | FI-Sii [22] | FR-LGt [22] | GL-NuF [22] |
| GL-ZaF [14] | IE-Cra [23] | PE-QFR [203] | RU-Che [14] | SE-Sto [22] | SJ-Adv [14] |
| UK-AMo [22] | US-ALQ [204] | US-Atq [14] | US-BZB [205] | US-BZF [206] | US-BZo [207] |
| US-EDN [208] | US-HB1 [209] | US-ICs [210] | US-Ivo [14] | US-KS3 [211] | US-Los [14] |
| US-Myb [212] | US-NC4 [213] | US-ORv [214] | US-OWC [215] | US-Srr [216] | US-StJ [217] |
| US-Tw1 [218] | US-Tw4 [219] | US-Tw5 [220] | US-WPT [221] | US-xBA [222] | |

**Woody Savannas (WSA)**

| AU-Ade [14] | AU-Gin [14] | AU-How [14] | AU-RDF [14] | BR-Npw [223] | ES-Cnd [23] |
|---|---|---|---|---|---|

Table 6: Meteorological Variables in CarbonSense

| Code | Description | Units |
|---|---|---|
| **Predictors** | | |
| TA_F | Air temperature | deg C |
| PA_F | Atmospheric pressure | kPa |
| P_F | Precipitation | mm |
| RH | Relative humidity | % |
| VPD_F | Vapor pressure deficit | hPa |
| WS_F | Wind speed | $\text{m s}^{-1}$ |
| USTAR | Wind shear | $\text{m s}^{-1}$ |
| WD | Wind direction | decimal degrees |
| NETRAD | Net radiation | $\text{W m}^{-2}$ |
| SW_IN_F | Incoming shortwave radiation | $\text{W m}^{-2}$ |
| SW_OUT | Outgoing shortwave radiation | $\text{W m}^{-2}$ |
| SW_DIF | Incoming diffuse shortwave radiation | $\text{W m}^{-2}$ |
| LW_IN_F | Incoming longwave radiation | $\text{W m}^{-2}$ |
| LW_OUT | Outgoing longwave radiation | $\text{W m}^{-2}$ |
| PPFD_IN | Incoming photosynthetic photon flux density | $\mu\text{mol Photon m}^{-2}\text{ s}^{-1}$ |
| PPFD_OUT | Outgoing photosynthetic photon flux density | $\mu\text{mol Photon m}^{-2}\text{ s}^{-1}$ |
| PPFD_DIF | Incoming diffuse photosynthetic photon flux density | $\mu\text{mol Photon m}^{-2}\text{ s}^{-1}$ |
| $CO_2$_F_MDS | $CO_2$ atmospheric concentration | $\mu\text{mol } CO_2\text{ mol}^{-1}$ |
| G_F_MDS | Soil heat flux | $\text{W m}^{-2}$ |
| LE_F_MDS | Latent heat flux | $\text{W m}^{-2}$ |
| H_F_MDS | Sensible heat flux | $\text{W m}^{-2}$ |
| **Targets** | | |
| NEE_VUT_REF | Net Ecosystem Exchange (variable USTAR) | $\mu\text{mol } CO_2\text{ m}^{-2}\text{ s}^{-1}$ |
| GPP_DT_VUT_REF | Gross Primary Production (daytime partitioning) | $\mu\text{mol } CO_2\text{ m}^{-2}\text{ s}^{-1}$ |
| GPP_NT_VUT_REF | Gross Primary Production (nighttime partitioning) | $\mu\text{mol } CO_2\text{ m}^{-2}\text{ s}^{-1}$ |
| RECO_DT_VUT_REF | Ecosystem Respiration (daytime partitioning) | $\mu\text{mol } CO_2\text{ m}^{-2}\text{ s}^{-1}$ |
| RECO_NT_VUT_REF | Ecosystem Respiration (nighttime partitioning) | $\mu\text{mol } CO_2\text{ m}^{-2}\text{ s}^{-1}$ |

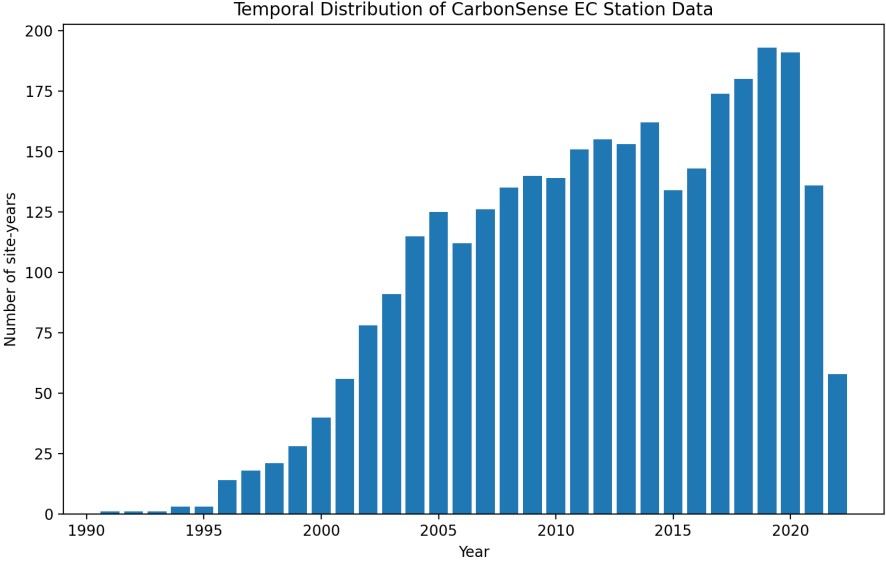

Figure 8: Temporal distribution of CarbonSense data. Data collected before 2000 does not have any MODIS data.

# B   EXPERIMENT DETAILS

## B.1   DATASET CONFIGURATION

As part of the CarbonSense pipeline, we filter out poorly gapfilled values. Each variable in the EC data has a corresponding "quality check" (QC) flag indicating if it was directly measured, gap filled during the ONEFlux pipeline (with varying gap fill quality levels), or simply taken from ERA5 reanalysis products. The tolerance level can be configured during the CarbonSense normalization process, and we discuss this further in the supplementary material.

We chose to use a maximum QC flag of 1, indicating all values in the dataset were either directly measured, or gap filled with high confidence. We found this had the best trade-off of data quality and quantity, as setting the maximum QC flag to 0 (only directly measured values) reduced the dataset size by 55%, while including medium-confidence values only increased it by 9%.

The data split was randomized within each ecosystem type. We held out 20% or 5 sites for each type, whichever is lower. The remaining sites were divided 80/20 between training and validation sets for EcoPerceiver, while our XGBoost model used all the training data for a cross-validation procedure.

## B.2   ECOPERCEIVER CONFIGURATION

Hyperparameter tuning for EcoPerceiver was performed with our train and validation splits, and comprised the bulk of the experiment efforts. Where possible, we started with our best guesses and ran a pseudo-random search based on intuition. A true random search of the parameter space would have been extremely sparse given the available compute resources.

We set our latent hidden size to 128, our input embedding size to 16, and the number of Fourier encoding frequencies to 12. This gave a total input hidden size of 40. Our context window is 32, meaning our model sees the previous 32 hours of observations. We use 8 WCA blocks followed by 4 CSA blocks. We set our observational dropout at 0.3 and use causal masking in all self-attention blocks. In keeping with [26], we employ weight sharing between all WCA blocks. With these hyperparameters our model weighs in at a very reasonable 988,633 parameters.

Heavier configurations were considered, but performance gains were minimal (see ablation studies) and the compute tradeoff made it impractical for anyone without multi-GPU cluster access to use the model. This is especially true for increasing the context window or latent hidden dimension.

## B.3   XGBOOST CONFIGURATION

Hyperparameters were found by random search. We used the same train/test split as the EcoPerceiver experiment; the train set was used in a 5-fold cross validation framework with 50 iterations. Once hyperparameters were found, we retrained XGBoost on all training data before running inference on the test set. Table 7 details the parameterization of our final model.

Since XGBoost is a tabular algorithm, we prepared geospatial data in a similar fashion to XBASE [4]; each spectral band represents a single input value to the model. The value is obtained by taking a weighted average of pixels based on Euclidean distance from the center of the image. Missing pixels were removed from this process, and the weights of the remaining pixels were increased to accommodate for this. The code for this procedure is provided with CarbonSense.

Table 7: XGBoost Hyperparameters

| Parameter | Value |
| --- | --- |
| learning_rate | 0.1 |
| alpha | 0.1 |
| gamma | 0.4 |
| lambda | 0.0 |
| max_depth | 9 |
| min_child_weight | 9 |
| n_estimators | 150 |
| subsample | 0.7 |
| scale_pos_weight | 0.5 |
| colsample_bytree | 0.7 |
| colsample_bylevel | 0.8 |

## B.4   REPRODUCIBILITY AND RELIABILITY

Both EcoPerceiver and XGBoost were trained with reproducibility in mind. Once optimal hyperparameters were found, we performed 10 experiments

with each model in order to obtain a reliable measure of performance (inspired by [224]). Set seeds were provided to all frameworks utilizing RNG, and distributed dataloader workers were also seed-controlled to ensure full reproducibility of our results.

The seeds for our experiments were simple integer values (0, 10, 20, ... , 90) and were provided for the final training runs after hyperparameters had already been chosen.

## B.5 DETAILED RESULTS

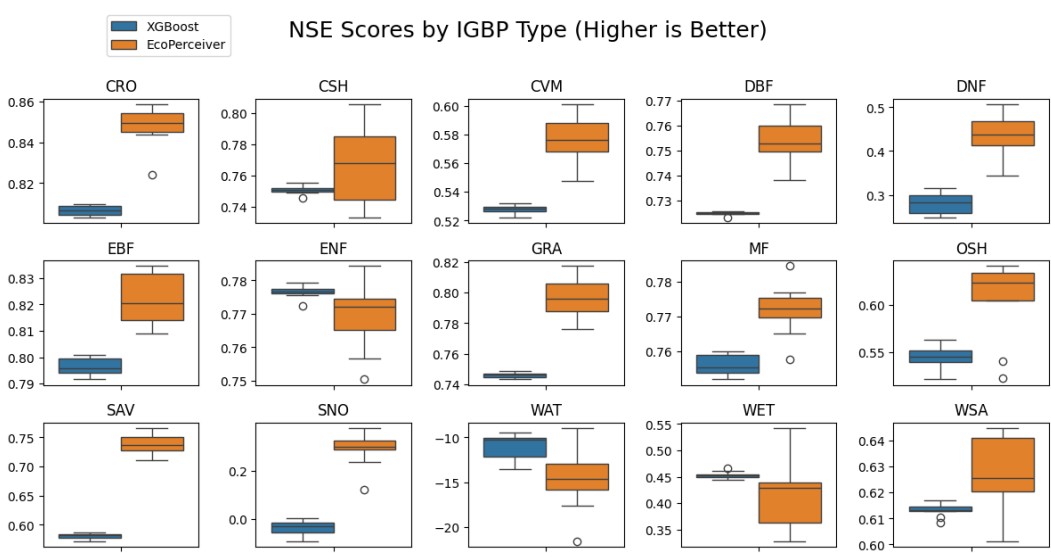

Figure 9: NSE scores of EcoPerceiver and XGBoost across different IGBP types. Each chart represents 10 experiments with different seeds. Whiskers indicate approximately 1.5 x interquartile range.

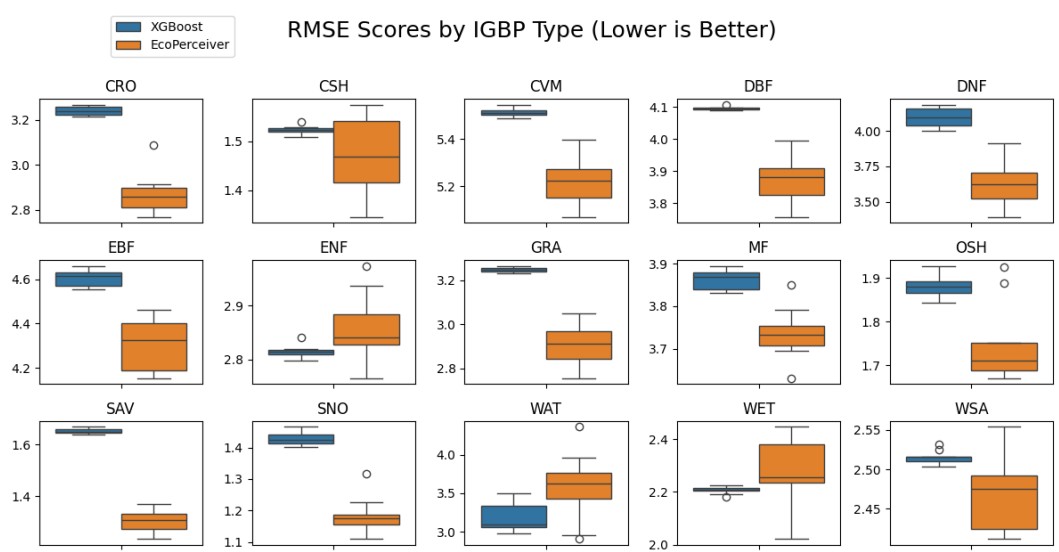

Figure 10: RMSE scores of EcoPerceiver and XGBoost across different IGBP types. Each chart represents 10 experiments with different seeds. Whiskers indicate approximately 1.5 x interquartile range.

Here we take a closer look at the performance of our models across different IGBP types. Figure 11a and Figure 11b show box plots of our models' test set performance using NSE and RMSE respectively, across all seeds. Note that the y-axis changes between each plot. We do this because the variance in model performance across different seeds was generally small, and charting all box plots on the same axis made the chart unreadable.

As discussed in Section 5, EcoPerceiver performs better than the XGBoost model in 12 out of 15 IGBP types. In 1 of the 3 that XGBoost wins, both models do substantially worse than simply guessing the mean, which makes the results for WAT challenging to interpret. The other 2, ENF and WET, are not significantly better than EcoPerceiver's performance and have mean NSE advantages of +0071 and +0.0393 respectively in favour of XGBoost. This could be explained by the nature of data splitting; once hyperparameters were obtained for XGBoost, it was able to train on the entirety of the train split, while EcoPerceiver still had to reserve 20% of the split for validation testing to measure convergence. Both ENF and WET had significant train set prevalence, so these represent IGBP types where XGBoost was most able to take advantage of additional data.

Imperfect hyperparameter selection could also account for the lack of consistent performance. While XGBoost is lightweight enough for a virtually exhaustive parameter search with cross validation, deep models have significantly higher experiment overhead. Due to compute limitations, we were limited in how thoroughly we could explore model configurations for EcoPerceiver.

### B.6   ADDITIONAL MODEL COMPARISONS

Our primary experiments with EcoPerceivercompare its performance with an XGBoost baseline, as this is the model we found performed the best out of the tabular architectures. Here we show the comparative performance of these models against additional baselines including a vanilla transformer, a random forest, and a simple linear regression model. The NSE and RMSE scores are shown in Tables 8 and 9 respectively. We kept these results partitioned by IGBP type to demonstrate the consistency of the results. Note that since linear regression models are deterministic, this model was not trained with multiple seeds to find an average performance.

These results further reinforce the idea that XGBoost remains a competitive baseline for DDCFM with non-deep learning methods. EcoPerceiver still demonstrates superior performance in the vast majority of IGBP types, but the vanilla transformer model notably wins out in WAT. This suggests that even without all the innovations of EcoPerceiver, deep learning models are highly effective at multimodal modelling in the context of DDCFM.

Table 8: NSE scores for all models.

| IGBP | Linear Model | Random Forest | XGBoost | Transformer | EcoPerceiver |
|------|------|------|------|------|------|
| CRO | 0.6315 | 0.7292 | 0.8066 | 0.8126 | **0.8482** |
| CSH | 0.5072 | 0.7107 | 0.7510 | 0.7381 | **0.7670** |
| CVM | 0.4282 | 0.5179 | 0.5277 | 0.4809 | **0.5763** |
| DBF | 0.5333 | 0.6875 | 0.7250 | 0.7318 | **0.7547** |
| DNF | 0.2178 | 0.2975 | 0.2803 | 0.2745 | **0.4336** |
| EBF | 0.6381 | 0.7938 | 0.7966 | 0.7464 | **0.8220** |
| ENF | 0.5934 | 0.7375 | **0.7765** | 0.7154 | 0.7694 |
| GRA | 0.6264 | 0.7258 | 0.7461 | 0.6803 | **0.7967** |
| MF | 0.6043 | 0.7250 | 0.7559 | 0.7316 | **0.7717** |
| OSH | 0.0585 | 0.4113 | 0.5451 | 0.5050 | **0.6060** |
| SAV | 0.1632 | 0.4174 | 0.5802 | 0.5288 | **0.7368** |
| SNO | -0.6223 | -0.0130 | -0.0370 | -0.1229 | **0.2898** |
| WAT | -32.6151 | -27.8940 | -11.0524 | **-9.6845** | -14.4010 |
| WET | 0.0976 | 0.2508 | **0.4530** | 0.4138 | 0.4137 |
| WSA | 0.4946 | 0.5575 | 0.6132 | 0.5560 | **0.6267** |

Table 9: RMSE scores for all models.

| IGBP | Linear Model | Random Forest | XGBoost | Transformer | EcoPerceiver |
|------|-------------|---------------|---------|-------------|--------------|
| CRO | 4.4698 | 3.8319 | 3.2381 | 3.1873 | **2.8677** |
| CSH | 2.1417 | 1.6411 | 1.5224 | 1.5613 | **1.4709** |
| CVM | 6.0688 | 5.5726 | 5.5157 | 5.7824 | **5.2236** |
| DBF | 5.3360 | 4.3661 | 4.0959 | 4.0451 | **3.8678** |
| DNF | 4.2721 | 4.0485 | 4.0974 | 4.1143 | **3.6322** |
| EBF | 6.1426 | 4.6365 | 4.6050 | 5.1420 | **4.3070** |
| ENF | 3.7959 | 3.0497 | **2.8141** | 3.1755 | 2.8579 |
| GRA | 3.9406 | 3.3759 | 3.2487 | 3.6451 | **2.9059** |
| MF | 4.9190 | 4.1002 | 3.8633 | 4.0511 | **3.7361** |
| OSH | 2.7043 | 2.1384 | 1.8796 | 1.9609 | **1.7475** |
| SAV | 2.3315 | 1.9455 | 1.6514 | 1.7497 | **1.3070** |
| SNO | 1.7876 | 1.4126 | 1.4291 | 1.4873 | **1.1816** |
| WAT | 5.3247 | 4.9366 | 3.1838 | **3.0019** | 3.5802 |
| WET | 2.8352 | 2.5834 | **2.2073** | 2.2851 | 2.2830 |
| WSA | 2.8752 | 2.6903 | 2.5153 | 2.6952 | **2.4706** |

## B.7 ABLATION STUDIES

Here we present ablation studies on EcoPerceiver. While we seek to provide a broad view of the impacts of our architectural decisions, note that running all these tests with a large number of seeds was computationally infeasible for us. Instead we ran each ablation test with a single seed ("00") and plotted performance below. The boxplots in this section therefore represent scores across different IGBP types.

To improve interpretability, the single WAT site was removed from the test set for these ablation studies. As discussed in Section 5.4, this is a one-shot EC station with ecosystem dynamics far out of distribution compared to other biomes. Model performance on this one site was exceedingly poor and variable, with NSE scores ranging from -3.0 to -20.2. Therefore, when running a single experiment per model, it would have a massively outsized and unpredictable effect on model scores.

### B.7.1 ARCHITECTURAL CHARACTERISTICS

Figure 11 shows a broad comparison of various architectural configurations on performance. We break down statistics for each architecture in Table 10. Each of these models was meant to interrogate the decisions we made when constructing the final EcoPerceiver. A summary of each model follows:

- **Ours**: This is the final version of EcoPerceiver, run with a single seed.

- **No Causal Masking**: A model where latent tokens are allowed to self-attend non-causally.

- **No Observational Dropout**: A model where all observations are kept during training, rather than randomly masking a portion of them during windowed cross-attention.

- **No Fourier Encoding**: A model with no Fourier encoding. This means the final input vector for cross-attention is of length $H_i = l_{emb} + 1$.

- **Gapfilled Data**: This ablation used our final model architecture, but used a version of CarbonSense where we keep the gapfilled values from the underlying ONEFlux pipeline rather than handling missing data within the model.

- **No Image Data**: This ablation also uses our final model, but does not pass in any image data during training or inference. It operates entirely on tabular data.

- **Vanilla Transformer**: A model without WCA or CSA. It is a transformer encoder as described in [225]. Each forward pass ingests the tabular and image data (tokenized as in our final model) for a single timestep. Regression is performed via linear probing on the latent space after 6 encoder layers. This is the best approximation of a vanilla transformer encoder in the context of our multimodal dataset, and it is the model used in the model comparison in Appendix B.6.

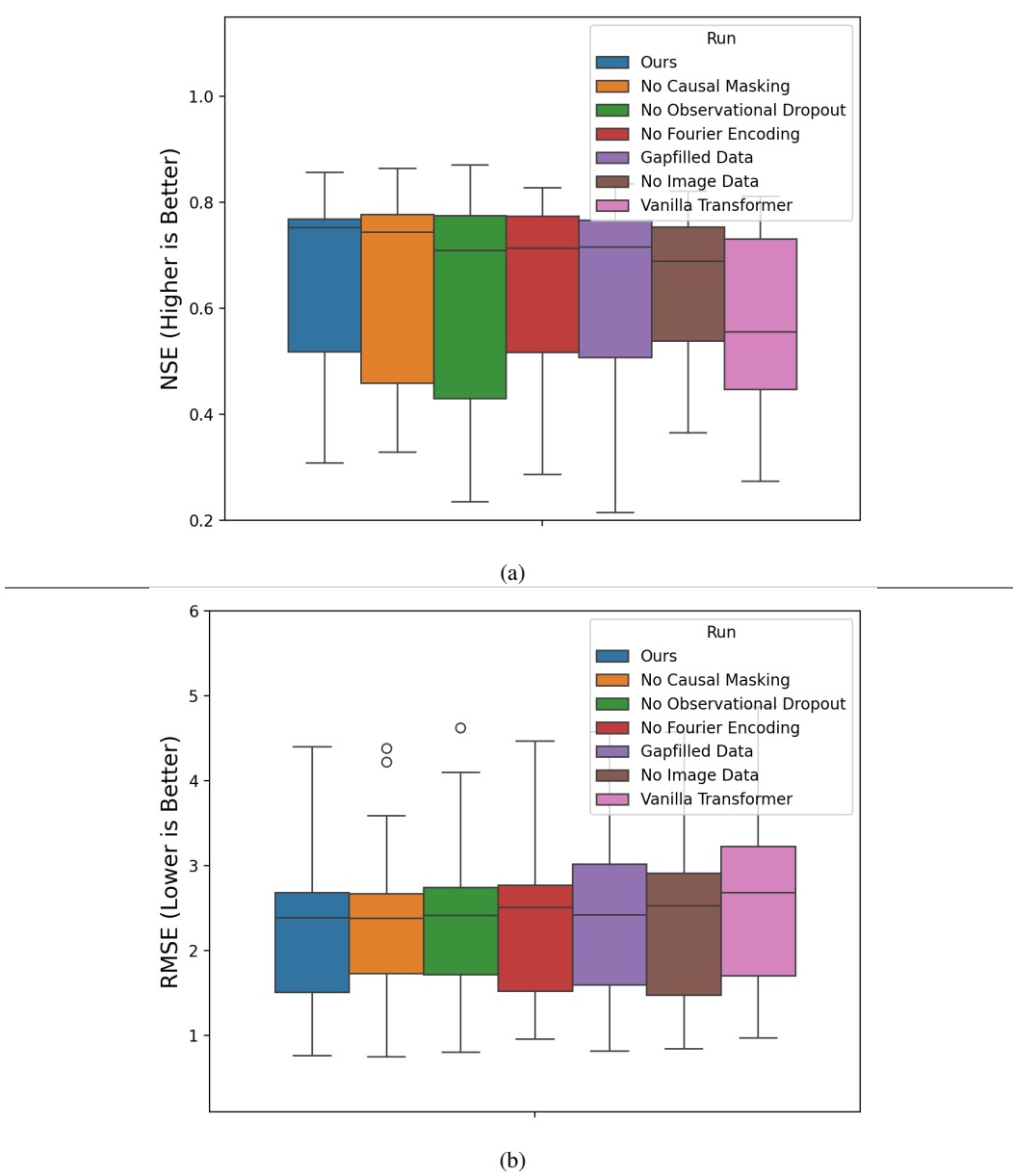

Figure 11: Boxplots of NSE (a) and RMSE (b) scores for ablation studies.

Our model performs better than all other ablation models broadly. The largest jump in performance is between the vanilla transformer and all other architectures. This indicates that WCA has a substantial effect on the ability of the model to predict carbon fluxes. It is unclear whether this benefit comes from the way WCA processes information and maps it to the latent space, or whether this is a reflection of the non-Markovian nature of ecosystem dynamics. Since the vanilla transformer only consumes data from the timestep for which it is predicting fluxes, it is not capturing the context window of observations, and designing a vanilla transformer to ingest a full context window would result in an intractable amount of processing for the forward pass.

The model without image data also underperformed, suggesting that satellite imagery provides substantial information with respect to carbon flux prediction. Additionally, the gapfilled data experiment shows that EcoPerceiver is able to process missing data in-model effectively. The remaining ablations (fourier encoding, observational dropout, causal masking) demonstrate that these architectural considerations may provide a modest benefit, though the results are far closer.

Table 10: NSE and RMSE by model ablation. *t_mean* and *t_std* represent truncated means and standard deviations to account for the outlier biome ("WAT").

| Model | NSE | | | | RMSE | | | |
|---|---|---|---|---|---|---|---|---|
| | t_mean | t_std | median | iqr | t_mean | t_std | median | iqr |
| Vanilla Transformer | 0.557 | 0.249 | 0.556 | 0.284 | 2.709 | 1.165 | 2.648 | 1.603 |
| No Image Data | 0.635 | 0.186 | 0.689 | 0.215 | 2.515 | 1.134 | 2.478 | 1.582 |
| Gapfilled Data | 0.635 | 0.199 | 0.717 | 0.259 | 2.474 | 1.077 | 2.376 | 1.492 |
| No Fourier Encoding | 0.624 | 0.254 | 0.715 | 0.257 | 2.456 | 1.043 | 2.481 | 1.098 |
| No Obs. Dropout | 0.634 | 0.196 | 0.710 | 0.345 | 2.468 | 1.091 | 2.339 | 1.105 |
| No Causal Masking | 0.654 | 0.171 | 0.744 | 0.318 | 2.428 | 1.077 | **2.252** | 0.836 |
| Ours | **0.669** | 0.166 | **0.753** | 0.251 | **2.382** | 1.055 | 2.313 | 1.284 |

In particular, the causal masking performed roughly equivalently to our final model. However, we decided to keep the causal masking in the final model anyway as it did not affect wall time, and contributed to the model's ecological validity.

### B.7.2 CONTEXT WINDOW LENGTH

We conducted further experiments to test the assumption that DDCFM benefits from ingesting data in a temporal context window. The results are shown in Figure 12. There is a clear performance advantage going from a context window of 8 hours to 16, and a small advantage going from 16 to 32. Anecdotally, this reflects our findings from early hyperparameter tuning experiments. Going from 32 to 64 did not meaningfully improve performance, but it did significantly increase our wall time and memory usage. We therefore used a context window of 32 for our main experiments.

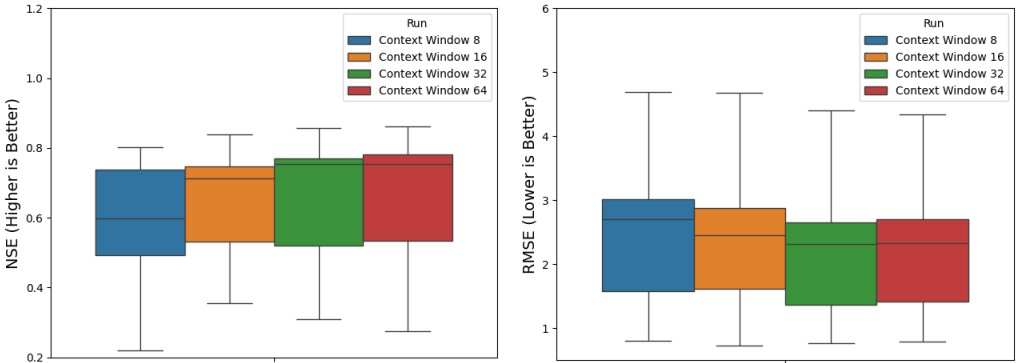

Figure 12: Effects of context window length on NSE (left) and RMSE (right) of test site inference for EcoPerceiver.

## B.8   ECOSYSTEM-SPECIFIC DDCFM

DDCFM is not always performed at the global scale; many research teams have studied it in the context of specific regions and ecosystem types [5, 6, 9, 12]. This use case is one of the reasons for CarbonSense's partitioned structure. As a proof of concept, we ran a single experiment with EcoPerceiver where we use the same parameters and train/test split as our main experiment, but only include the DBF sites. We then compared the test set performance against our main model. Table 11 shows the results - our model trained on multiple ecosystem types had notably better performance despite a similar convergence time.

Table 11: EcoPerceiver performance on DBF sites when trained on only DBF data vs trained on all sites

| Only DBF | | All Sites | |
| --- | --- | --- | --- |
| NSE | RMSE | NSE | RMSE |
| 0.7405 | 3.9782 | **0.7532** | **3.8806** |

As with the ablation studies, we did not have the compute resources to do 10 seeds for every experiment variation - but it shows the flexibility of CarbonSense for different research scenarios. It also provides preliminary evidence that DDCFM with multimodal models may benefit from adding more training data even if it is relatively out of distribution.

## B.9   QUALITATIVE ANALYSIS

While error metrics are useful for assessing the aggregate performance of the model, we encourage researchers to inspect the model outputs in comparison to the observed data. As an example, consider Figures 13 and 14 below. Both of these were randomly selected 4-day stretches of data from their respective sites. Both models appear able to model GF-Guy very well, but not CA-LP1, and this may be counterintuitive at first glance. But there's quite a bit going on here.

GF-Guy is an evergreen broadleaf forest in the tropics. This is not highly prevalent type in Carbon-Sense, but its carbon fluxes appear quite stable from day to day. We found that ecosystems with this interseasonal stability tend to be more easily modelled in our experiments, though this is not an easy metric to quantify. It should be noted that the y-axis has a much larger scale, so while the models appear close to the ground truth, they often have an error in excess of 5 μmol $CO_2$ m$^{-2}$. This again highlights the importance of using NSE as an error metric - RMSE will unfairly punish highly active ecosystems like this due to higher natural variance in carbon fluxes.

CA-LP1 is an evergreen needleleaf forest in the temperate region, which is one of the best represented ecosystems in CarbonSense, yet both models struggle with it (despite low *absolute* error), especially in the winter. Reading into this site reveals it is a pine beetle-attacked forest [97]; disturbances like these can be challenging to model as we discussed in 2. This gives future work in DDCFM a vector for potential improvement.

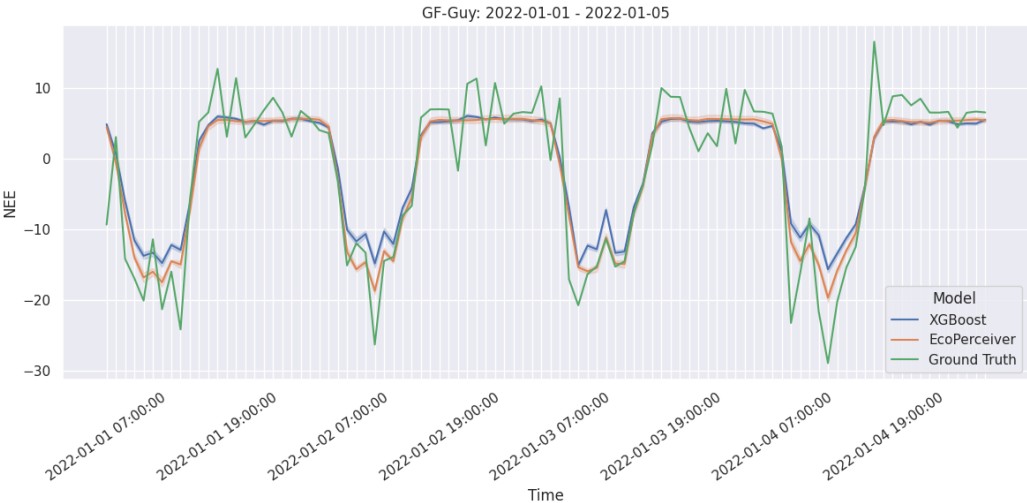

Figure 13: Hourly data and model results for GF-Guy, an evergreen broadleaf forest station in French Guiana.

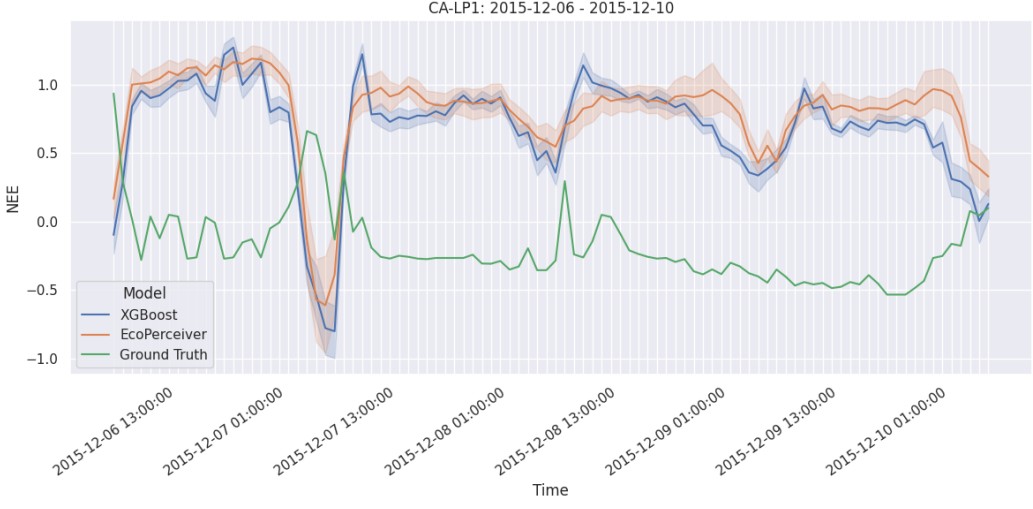

Figure 14: Hourly data and model results for CA-LP1, a pine beetle-attacked evergreen needleleaf forest in northern British Columbia.

