# CarbonSense Data Card

| **CarbonSense** | **Dataset Summary** |
|---|---|
| **This doc**: [GitHub Link](GitHub Link)

**Dataset**: [Zenodo Link](Zenodo Link)

**Data Pipeline Code**: [GitHub Link](GitHub Link)

**Data Card Author**: Matthew Fortier

**Correspondence**: matthew.fortier@mila.quebec | This data card describes CarbonSense, a multimodal dataset for carbon flux modelling.

CarbonSense integrates biophysical data collected from field observation towers with satellite imagery of the surrounding geography. The primary use for CarbonSense is to use the data to predict a carbon flux value for that area at a given time, indicating how much carbon is being absorbed or released by the surrounding ecosystem. |

## Dataset Authors

| Author | Affiliation(s) | Funding |
|---|---|---|
| Matthew Fortier (Owner) | Mila Quebec AI Institute Polytechnique Montréal | IVADO AI, Biodiversity and Climate Change Initiative [[link](link)] |
| Mats L. Richter (Contributor) | ServiceNow | |
| Oliver Sonnentag (Contributor) | Université de Montréal | |
| Chris Pal (Contributor) | Mila Quebec AI Institute Polytechnique Montréal | |

## Dataset Snapshot

| | |
|---|---|
| **Data modality** | Multimodal (tabular, image) |
| **Size of dataset** | 12.4 GB |
| **Number of instances** | 26,973,576 |
| **Number of fields per instance** | 36 [30 predictors + 5 targets + index] |

# Motivations & Use

| Key Domain Applications | Motivating Factors | Intended Use | License |
|---|---|---|---|
| Machine Learning, Deep Learning, Multimodal Learning, Carbon Dynamics, Climate Change | Providing a shared dataset for deep learning research into carbon flux modelling. | Safe for research use. Production use should be approached with caution due to data imbalances. | CarbonSense is provided under Creative Commons 4.0 license, meaning anyone is free to remix, adapt, and redistribute given proper attribution, even commercially. |

| Citation | Access and Maintenance |
|---|---|
| @misc{fortier2024carbonsense,
    title={CarbonSense: A Multimodal Dataset
        and Baseline for Carbon Flux Modelling},
    author={Matthew Fortier and Mats L. Richter
        and Oliver Sonnentag and Chris Pal},
    year={2024},
    eprint={2406.04940},
    archivePrefix={arXiv},
    primaryClass={cs.LG}
} | CarbonSense is open access and can be downloaded from Zenodo here.

Zenodo is a CERN-funded initiative with a robust longevity plan. The data will be available and versioned for at least 20 years. |

# Dataset Structure

| Overview | Dataset is divided into geographic sites (385 total). Each site will have one or more observation windows (ex: 2005-01-01_2016-12-31). Each observation window will have four resources: metadata, predictors, images, and targets. |
|---|---|
| Directory Structure | {site}/{window}/{resource} |
| Example File Path | CA-LP1/2007-01-01_2020-12-13/targets.csv |

## Resources

| meta.json | Contains site metadata such as ecosystem type and geographic location |
|---|---|
| predictors.csv | Meteorological and environmental variables at hourly intervals - also referred to as eddy covariance (EC) data |
| targets.csv | Carbon fluxes at hourly intervals |
| modis.pkl | Satellite data of surrounding area at daily intervals - also referred to as MODIS data |

## Dataset Fields

| Resource | Field | Description | Type | Null ratio |
|---|---|---|---|---|
| predictors.csv | timestamp | Pandas datetime object used for indexing | string | 0.000 |
| | DOY | Day of year | float | 0.000 |
| | TOD | Time of day | float | 0.000 |
| | TA_F | Air temperature | float | 0.115 |
| | SW_IN_F | Incoming shortwave radiation | float | 0.123 |
| | LW_IN_F | Incoming longwave radiation | float | 0.387 |
| | VPD_F | Vapor pressure deficit | float | 0.147 |
| | PA_F | Air pressure | float | 0.264 |
| | P_F | Precipitation | float | 0.263 |
| | WS_F | Wind speed | float | 0.193 |
| | WD | Wind direction | float | 0.190 |
| | RH | Relative humidity | float | 0.166 |
| | USTAR | Wind friction velocity | float | 0.228 |
| | NETRAD | Net radiation | float | 0.241 |
| | PPFD_IN | Incoming photosynthetic photon flux density | float | 0.262 |
| | PPFD_OUT | Outgoing photosynthetic photon flux density | float | 0.584 |
| | SW_OUT | Outgoing shortwave radiation | float | 0.351 |
| | LW_OUT | Outgoing longwave radiation | float | 0.431 |
| | CO2_F_MDS | CO2 concentration | float | 0.185 |
| | G_F_MDS | Soil heat flux | float | 0.371 |
| | LE_F_MDS | Latent heat flux | float | 0.178 |
| | H_F_MDS | Sensible heat flux | float | 0.159 |

| targets.csv | timestamp | Pandas datetime object used for indexing | string | 0.000 |
|---|---|---|---|---|
| | NEE_VUT_REF | Net ecosystem exchange | float | 0.230 |
| | GPP_DT_VUT_REF | Gross primary production, daytime partition | float | 0.040 |
| | GPP_NT_VUT_REF | Gross primary production, nighttime partition | float | 0.039 |
| | RECO_DT_VUT_REF | Ecosystem respiration, daytime partition | float | 0.040 |
| | RECO_NT_VUT_REF | Ecosystem respiration, nighttime partition | float | 0.039 |
| meta.json | SITE_ID | Eddy covariance (EC) station site identifier | string | 0.000 |
| | LOCATION_LAT | EC station latitude | float | 0.000 |
| | LOCATION_LON | EC station longitude | float | 0.000 |
| | LOCATION_ELEV | EC station elevation above sea level | float | 0.136 |
| | IGBP | IGBP land cover type (ecosystem class) | string | 0.000 |
| | SOURCES | Source datasets used for this site, comma-separated | string | 0.000 |
| | TIME | Temporal window covered by data | list | 0.000 |
| modis.pkl | TIMESTAMP* | Pandas datetime object used as key value for indexing images | string | N/A |
| | dim0** | MCD43A4 Band 1 | array | N/A |
| | dim1** | MCD43A4 Band 2 | array | N/A |
| | dim2** | MCD43A4 Band 3 | array | N/A |
| | dim3** | MCD43A4 Band 4 | array | N/A |
| | dim4** | MCD43A4 Band 5 | array | N/A |
| | dim5** | MCD43A4 Band 6 | array | N/A |
| | dim6** | MCD43A4 Band 7 | array | N/A |

| | dim7** | MCD43A2 Band 1 | array | N/A |
|---|---|---|---|---|
| | dim8** | MCD43A2 Band 2 | array | N/A |

\* The modis.pkl resource contains a dictionary of key-value pairs. The keys are timestamps corresponding to satellite observations, typically once per day. As such, there are no inherently null values; if a satellite image is not present at a given time, there will simply be no key for it

\*\* The values of the dictionary are 9-channel satellite images of the associated EC station.

# Data Collection & Processing

## Eddy Covariance Data

| Overview | Collection |
|---|---|
| Eddy covariance (EC) data is the tabular data used in CarbonSense. It is made up of environmental and meteorological predictors, as well as carbon flux targets. | Raw data was aggregated from existing datasets:

● FLUXNET 2015
● Ameriflux
● ICOS 2023
● ICOS WW

These regional flux networks release data at various intervals; ICOS releases updated datasets multiple times per year, FLUXNET still has not released a second version since 2020.

The criteria for inclusion are data sources which use the ONEFlux processing pipeline; this ensures consistent variables and units across EC stations.

**Collection Cadence**: Static
**Collection Date**: 2024-03-01 |

**Processing**

- Hourly data extracted from releases. Half-hourly data downsampled to hourly
- Relevant columns extracted (all columns in *predictors.csv* and *targets.csv* above)
  - Corresponding quality check columns also kept
- Sentinel missing values deleted
- Data from source collections fused and partitioned by EC station
  - Stations present in multiple sources are combined, favouring values from more recent release in the event of collisions
  - Each station will have one or more subdirectories of contiguous data coverage. Separating by contiguity makes data easier to handle for time series modelling. Example:

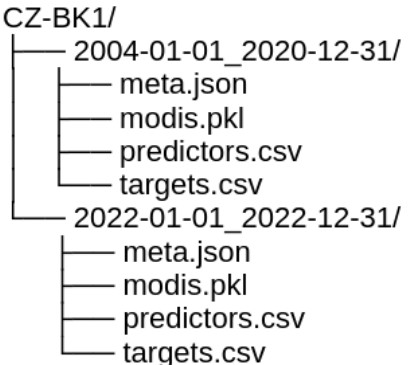

```
CZ-BK1/
├── 2004-01-01_2020-12-31/
│   ├── meta.json
│   ├── modis.pkl
│   ├── predictors.csv
│   └── targets.csv
└── 2022-01-01_2022-12-31/
    ├── meta.json
    ├── modis.pkl
    ├── predictors.csv
    └── targets.csv
```

- EC station geography and temporal coverage extracted, used as metadata for satellite data collection
- Min-max normalization procedure
  - Min/max values were determined for each variable
    - Examples
      - Day of year [0, 366]
      - Air temperature [-80, 80] (degrees celsius)
      - Wind speed [0, 100] (meters per second)
    - These were chosen by objective bounds, or by generous feasible bounds based on historical extremes (for example, highest recorded wind speed is 103 m/s)
  - Variables were mapped from min-max values to [-1, 1) for cyclic variables, and [-0.5, 0.5) for acyclic variables
    - Cyclic variables are like wind direction and time of day, where the maximum and minimum values are identical
    - This distinction makes it easy for models to understand the variables under Fourier feature encoding, which our baseline model uses
    - Our data pipeline allows for compiling CarbonSense without these distinctions via config file
- Data filtering
  - Quality check flags are used to identify gap-filled data points
  - Data with a quality check greater than or equal to 2 ("moderate confidence gapfilling") are discarded
    - The threshold for this procedure can be controlled with the pipeline config file
  - Quality check flags are then removed
- Data are then split into *predictors.csv* and *targets.csv* files, with all carbon flux values going into *targets.csv*. Timestamp variable remains in both files for indexing purposes.
- Site metadata are placed in individual *meta.json* files and stored alongside predictors and targets.

| Satellite Data | |
|---|---|
| **Overview**

CarbonSense uses Moderate Resolution Imaging Spectroradiometer (MODIS) satellite data products. The MCD43A4 and MCD43A2 are the only products in use in v1.0.0 of the dataset. | **Collection**

Satellite data was acquired through Google Earth Engine (GEE). A notebook in the GitHub repository is used to access GEE through Google Colab.

**Collection Cadence**: Static
**Collection Date**: 2024-03-01 |

**Processing**

- Metadata generated from EC data pipeline is used to specify location / time of satellite data to GEE
- 7 reflectance bands are used from MCD43A4; the rest are discarded
- 2 bands (water and snow cover) are used from MCD43A2; the rest are discarded
- MCD43A4 data is cleaned and normalized
    - Reflectance values in MCD43A4 range from 0 to 10,000 with a fill value of 32767
    - We map [0, 10000) -> [0, 1.0) and change fill values from (32767) -> (-1)
        - This makes it relatively easy for ingestion by neural network
- MCD43A2 data is cleaned
    - Both bands contain categorical data
    - Snow cover band
        - 1 = snow, 0 = no snow
        - 255 = fill value -> we remap to -1
    - Water cover band
        - Shallow ocean, shallow inland water, ephemeral water, deep inland water, moderate or continental ocean, deep ocean -> mapped to 1 ('water')
        - Land, ocean coastlines and lake shorelines -> mapped to 0 ('no water')
        - 255 = fill value -> we remap to 0 (treated as 'no water' for simplicity, this is essentially just median gap filling as the vast majority of pixels are 'no water')
- MCD43A2 is stacked on top of MCD43A4 so that band 8 is snow, and band 9 is water
- Images with > 50% fill values are discarded
- Images are placed in dictionaries with the image timestamp being the key. These dictionaries are written as binary files labelled *modis.pkl* alongside EC data files.