# OpenReview forum: "CarbonSense: A Multimodal Dataset and Baseline for Carbon Flux Modelling"
_ICLR.cc/2025/Conference — ICLR 2025 Poster_

### Official Review · Reviewer_8jPr · 2024-10-31

**Soundness:** 3
**Presentation:** 3
**Contribution:** 3
**Rating:** 6
**Confidence:** 5

**Summary:**

This paper presents a new benchmark dataset, Carbon Sense, for predicting carbon flux from geospatial and meteorological data. Two models are evaluated on the new benchmark: 1) an XGBoost model which emulates the SOTA method from prior work, and 2) the EcoPerceiver model proposed in this paper. Domain-relevant evaluation metrics of RMSE and NSE (Nash-Sutcliffe Modelling Efficiency) are reported for each model and broken down by ecosystem type, with statistical tests of significance of the differences in performance. EcoPerceiver significantly outperforms the XGBoost model, indicating there is room for improvement on the benchmark relative to the previous SOTA with more sophisticated/tailored ML approaches.

**Strengths:**

- The benchmark has multimodal inputs, which helps fill a notable gap in multimodal benchmarks for remote sensing/geospatial ML.
- The paper presents useful and approachable background about the carbon flux modeling problem.
- The benchmark code is designed to allow flexibility in reproducing and extending or modifying the dataset based on user needs.
- The benchmark has a permissive CC-BY license.
- The EcoPerceiver method is well motivated based on the domain-specific carbon flux modeling problem.
- Domain relevant metrics are used for evaluation.

**Weaknesses:**

- The train/test splits are divided by station location, which avoids spatial autocorrelation issues. It seems there could be significant temporal autocorrelation within each split since there are many measurements from the same location. Is temporal autocorrelation a concern?
- The experiments only compared two models, XGBoost and EcoPerceiver. It would be useful to see additional models benchmarked (especially deep learning models) to get a sense of the variation in performance existing solutions on this benchmark.
- What does the purple vector in Figure 4 represent?

**Questions:**

- The train/test splits are divided by station location, which avoids spatial autocorrelation issues. It seems there could be significant temporal autocorrelation within each split since there are many measurements from the same location. Is temporal autocorrelation a concern?

- What does the purple vector in Figure 4 represent?

- How do other deep learning models (existing models besides the new EcoPerceiver) perform on CarbonSense?

---

> ### Author Response · Authors · 2024-11-23
> **Response to Reviewer 8jPr**
>
> We would like to thank the reviewer for their detailed and insightful comments. We've addressed them pointwise below.
>
> #### 1. **Temporal Autocorrelation:**
>
> The reviewer raises a very good point regarding temporal autocorrelation. If we understand correctly, the reviewer is referring to how different towers in the training set experience similar climatic conditions over time. If so, this is a valid observation; the model would have difficulty generalizing to a timeframe say 20 years in the future where the climate is quite different.
>
> However, DDCFM is typically used to upscale carbon fluxes within the temporal period of the training data, as opposed to projecting future fluxes (which is more in the realm of process-based modelling). Very good discussion point though, and we may add to the limitations section that the model should not be used to predict fluxes using inputs taken long after the training data ends.
>
> #### 2. **Comparison to Additional Models:**
>
> We acknowledge the reviewer’s suggestion to include additional models for comparison. Several reviewers have made similar requests, and we address this by running experiments with additional models. For deep learning models, we already provide a comparison to a vanilla transformer model in our ablation studies in Appendix B.6.
>
> For tabular models, we have taken this week to run additional experiments using a random forest as well as a simple linear regression model to give a better idea of the comparative gains in performance. The tables for NSE and RMSE can be found below, and we will include them in the final version of the manuscript.
>
> #### 3. **Interpretation of the Purple Vectors in Figure 4:**
>
> We apologize for the confusion regarding the purple vectors in Figure 4. These vectors represent the latent arrays of the EcoPerceiver model. A more detailed view of these latent arrays can be seen in Figure 6, but we agree with the reviewer that the purple vectors should be labeled for clarity in Figure 4. This will be corrected in the final version of the manuscript.
>
> #### NSE Table
>
> | IGBP | Linear Model | Random Forest | XGBoost    | Transformer | EcoPerceiver |
> |------|--------------|---------------|------------|-------------|--------------|
> | CRO  | 0.6315       | 0.7292        | 0.8066     | 0.8126      | **0.8482**   |
> | CSH  | 0.5072       | 0.7107        | 0.7510     | 0.7381      | **0.7670**   |
> | CVM  | 0.4282       | 0.5179        | 0.5277     | 0.4809      | **0.5763**   |
> | DBF  | 0.5333       | 0.6875        | 0.7250     | 0.7318      | **0.7547**   |
> | DNF  | 0.2178       | 0.2975        | 0.2803     | 0.2745      | **0.4336**   |
> | EBF  | 0.6381       | 0.7938        | 0.7966     | 0.7464      | **0.8220**   |
> | ENF  | 0.5934       | 0.7375        | **0.7765** | 0.7154      | 0.7694       |
> | GRA  | 0.6264       | 0.7258        | 0.7461     | 0.6803      | **0.7967**   |
> | MF   | 0.6043       | 0.7250        | 0.7559     | 0.7316      | **0.7717**   |
> | OSH  | 0.0585       | 0.4113        | 0.5451     | 0.5050      | **0.6060**   |
> | SAV  | 0.1632       | 0.4174        | 0.5802     | 0.5288      | **0.7368**   |
> | SNO  | -0.6223      | -0.0130       | -0.0370    | -0.1229     | **0.2898**   |
> | WAT  | -32.6151     | -27.8940      | -11.0524   | **-9.6845** | -14.4010     |
> | WET  | 0.0976       | 0.2508        | **0.4530** | 0.4138      | 0.4137       |
> | WSA  | 0.4946       | 0.5575        | 0.6132     | 0.5560      | **0.6267**   |
>
>
> #### RMSE Table
>
> | IGBP | Linear Model | Random Forest | XGBoost    | Transformer | EcoPerceiver |
> |------|--------------|---------------|------------|-------------|--------------|
> | CRO  | 4.4698       | 3.8319        | 3.2381     | 3.1873      | **2.8677**   |
> | CSH  | 2.1417       | 1.6411        | 1.5224     | 1.5613      | **1.4709**   |
> | CVM  | 6.0688       | 5.5726        | 5.5157     | 5.7824      | **5.2236**   |
> | DBF  | 5.3360       | 4.3661        | 4.0959     | 4.0451      | **3.8678**   |
> | DNF  | 4.2721       | 4.0485        | 4.0974     | 4.1143      | **3.6322**   |
> | EBF  | 6.1426       | 4.6365        | 4.6050     | 5.1420      | **4.3070**   |
> | ENF  | 3.7959       | 3.0497        | **2.8141** | 3.1755      | 2.8579       |
> | GRA  | 3.9406       | 3.3759        | 3.2487     | 3.6451      | **2.9059**   |
> | MF   | 4.9190       | 4.1002        | 3.8633     | 4.0511      | **3.7361**   |
> | OSH  | 2.7043       | 2.1384        | 1.8796     | 1.9609      | **1.7475**   |
> | SAV  | 2.3315       | 1.9455        | 1.6514     | 1.7497      | **1.3070**   |
> | SNO  | 1.7876       | 1.4126        | 1.4291     | 1.4873      | **1.1816**   |
> | WAT  | 5.3247       | 4.9366        | 3.1838     | **3.0019**  | 3.5802       |
> | WET  | 2.8352       | 2.5834        | **2.2073** | 2.2851      | 2.2830       |
> | WSA  | 2.8752       | 2.6903        | 2.5153     | 2.6952      | **2.4706**   |

---

> > ### Comment · Reviewer_8jPr · 2024-11-26
> >
> > I appreciate the addition of more model comparisons, but these are still quite simple models - my question was about other deep learning models. I'm assuming that the expectation is that future methods developed to improve on this benchmark will be deep learning variants like EcoPerceiver and the vanilla Transformer. What other deep learning models would be sensible to compare here, e.g. an MLP or CNN?

---

> > > ### Author Response · Authors · 2024-12-02
> > > **Response to Reviewer 8jPr**
> > >
> > > We completely agree that testing more deep learning models on this problem would be really beneficial for the community, this is in fact exactly why we have formulated our work as consisting of two key contributions, (1) creating a benchmark, and (2) creating an initial deep learning based model which outperforms both the current state of the art techniques, and a simpler deep learning baseline (a vanilla transformer).
> > >
> > > The multimodality and sparsity of the data make it challenging to model quickly and easily with an off-the-shelf deep learning solution such as a standard CNN, so innovative new architectures are required and we truly hope that the existence of this benchmark will encourage many more deep learning researchers to develop solutions to this important problem.
> > >
> > > Our motivation with this work was to spark more interest in this problem domain by making it easier for the deep learning community to use this kind of data and by showing that a deep learning approach with non-trivial technical innovation is indeed capable of pushing the state of the art in modelling this problem. If you agree that it would be good to have more people in the ICLR community attacking this problem with more models using our benchmark, we hope that you will consider increasing your score so that this work is more likely to be accepted and gain more visibility by being presented at the conference.

---

### Official Review · Reviewer_ZVzD · 2024-11-03

**Soundness:** 3
**Presentation:** 4
**Contribution:** 3
**Rating:** 6
**Confidence:** 3

**Summary:**

Authors present CarbonSense as a machine learding ready multimodal dataset for carbon flux modeling. The dataset includes meteorological variables, MODIS 81 pixels satellite observations, and eddy covariance of carbon fluxes from hundreds of locations globally, Authors, in addition to the proposed dataset, presented a baseline machine learning model to predict carbon fluxes. The model shows the power of the multimodality of the dataset to improve canbon flux modeling performance.

**Strengths:**

* Paper is well written and cohesive. It is easy to follow and understand even for non-experts in the field
* Adds a clear contribution to existing dataset in terms of scale and modalities added. This will definitely help progress in the field.
* Dataset will be open for anyone to use. This is important for it to make any impact.

**Weaknesses:**

* The satellite imagery added is very low resolution wich limits it's potential usefulness.
* The dataset does not include many observations outside developed countries. Nothing the authors can do sinse they leverage existing EC stations available.

**Questions:**

* Consider adding satellite date from Sentinel 2 and 2 or Landsat which are higher resilution for days available. These datasets are open for anyone to use.
* Is it possible to keep expanding the dataset in a systematic way by ingesting EC and MODIS data periodically?

---

> ### Author Response · Authors · 2024-11-23
> **Response to Reviewer ZVzD**
>
> We thank the reviewer for their feedback and suggestions. We have addressed the concerns pointwise below.
>
> #### 1. **Satellite Imagery Resolution:**
>
> We acknowledge the reviewer’s concern about the low spatial resolution of MODIS imagery. This is the tradeoff we made in order to have more frequent images; since the model ingests 36 hours of data in a forward pass, it generally receives 1-2 MODIS images per pass, allowing the model to track changes more consistently.
>
> Sentinel 2 has a temporal resolution of ~5 days, and Landsat ~16 days, so we excluded them for now as most samples would not contain this imagery, and it would severely increase the dataset footprint. This also does not account for cloud cover which further restricts usable imagery.
>
> However, as part of a follow-up study, we are planning to expand the dataset by adding Phenocam data which is collected every 30 minutes at ~100 of the sites included in CarbonSense. This is a large undertaking, so we anticipate releasing this feature in a future version (V2) of CarbonSense.
>
> #### 2. **Periodic Expansion of the Dataset:**
>
> We also appreciate the suggestion to periodically expand the dataset by ingesting new EC and MODIS data. This is an idea we are considering, as it could turn CarbonSense into a continually improving resource. However, implementing this would also require buy-in from the underlying networks (Ameriflux / FLUXNET / ICOS), and a large amount of funding to ensure the continued support of necessary infrastructure. We have recently been approved for grants to extend this project, so continual integration of new data is being considered as a possible direction.

---

> > ### Comment · Reviewer_ZVzD · 2024-11-25
> > **Response to authors**
> >
> > Thank you for your response! PhenoCam would be great add and I understand periodic expansion should be something for the future.

---

> > > ### Author Response · Authors · 2024-12-02
> > > **Response to Reviewer ZVzD**
> > >
> > > Our goal is ultimately to provide as much data as we can so that future researchers will be able to pick subsets of data appropriate to their application / model architecture. Given this, we were wondering if you might consider revising your score to reflect the positive aspects of our contributions; we feel that getting CarbonSense accepted will go a long way toward exposing the deep learning research community to more climate-focused domain applications.

---

### Official Review · Reviewer_yJf4 · 2024-11-04

**Soundness:** 2
**Presentation:** 3
**Contribution:** 2
**Rating:** 6
**Confidence:** 5

**Summary:**

This paper proposes a standardized dataset called CarbonSense, which is a dataset compiled from various sources. The compilation steps include fusing the data together on the same hourly scale, extracting relevant features, and doing min max normalization. Along with the dataset, the paper also proposes a data driven model based on Perceiver called EcoPerceiver. This proposed model uses transformer architecture to cross attend all input features. The paper also implemented a SOTA approach baseline and compared its performance on the proposed dataset with EcoPerceiver.

**Strengths:**

CarbonSense integrates diverse data modalities—measured carbon fluxes, meteorological predictors, and satellite imagery—across a wide array of ecosystems. Researchers can use this dataset as a standardized benchmark.

**Weaknesses:**

1.The dataset was compiled from multiple sources with various modalities, which may introduce inconsistency or OOD samples when doing model training. Careful data analysis can be helpful
2. The experiment shows the proposed EcoPerceiver outperformed the current SOTA approach for most IGBP types especially WET, WAT, and ENF. However, the paper did not include an ablation study to showcase why the proposed model achieved this performance.
3.There is only one baseline compared and there is no model with single modalities.

**Questions:**

How does the Perceiver model compare to other advanced machine learning models in terms of performance? In addition, which component of EcoPerceiver affects performance the most? I think adding more machine learning model baselines and conducting ablation study would help determine the model's relative strengths and weaknesses.

---

> ### Author Response · Authors · 2024-11-23
> **Response to Reviewer yjf4**
>
> We would like to thank the reviewer for their valuable feedback. Below, we provide clarifications and additional information to address the points raised.
>
> #### 1. **Ablation Study:**
>
> We would like to clarify that we do provide extensive ablation studies in Appendix B.6 of the manuscript, where we compare model performance with various architectural components removed, and experiment with different context window lengths.
>
> #### 2. **Comparison to Additional Baselines:**
>
> We agree with the reviewer that comparing EcoPerceiver to additional baseline models could further strengthen the study. In the ablation section, we compare EcoPerceiver to a vanilla transformer model, as we believe this comparison is the most relevant given the data sparsity and multimodality.
>
> For tabular models, we have taken this week to run additional experiments using a random forest as well as a simple linear regression model to give a better idea of the comparative gains in performance. The tables for NSE and RMSE can be found below, and we will include them in the final version of the manuscript.
>
> #### NSE
>
> | IGBP | Linear Model | Random Forest | XGBoost    | Transformer | EcoPerceiver |
> |------|--------------|---------------|------------|-------------|--------------|
> | CRO  | 0.6315       | 0.7292        | 0.8066     | 0.8126      | **0.8482**   |
> | CSH  | 0.5072       | 0.7107        | 0.7510     | 0.7381      | **0.7670**   |
> | CVM  | 0.4282       | 0.5179        | 0.5277     | 0.4809      | **0.5763**   |
> | DBF  | 0.5333       | 0.6875        | 0.7250     | 0.7318      | **0.7547**   |
> | DNF  | 0.2178       | 0.2975        | 0.2803     | 0.2745      | **0.4336**   |
> | EBF  | 0.6381       | 0.7938        | 0.7966     | 0.7464      | **0.8220**   |
> | ENF  | 0.5934       | 0.7375        | **0.7765** | 0.7154      | 0.7694       |
> | GRA  | 0.6264       | 0.7258        | 0.7461     | 0.6803      | **0.7967**   |
> | MF   | 0.6043       | 0.7250        | 0.7559     | 0.7316      | **0.7717**   |
> | OSH  | 0.0585       | 0.4113        | 0.5451     | 0.5050      | **0.6060**   |
> | SAV  | 0.1632       | 0.4174        | 0.5802     | 0.5288      | **0.7368**   |
> | SNO  | -0.6223      | -0.0130       | -0.0370    | -0.1229     | **0.2898**   |
> | WAT  | -32.6151     | -27.8940      | -11.0524   | **-9.6845** | -14.4010     |
> | WET  | 0.0976       | 0.2508        | **0.4530** | 0.4138      | 0.4137       |
> | WSA  | 0.4946       | 0.5575        | 0.6132     | 0.5560      | **0.6267**   |
>
>
> #### RMSE
>
> | IGBP | Linear Model | Random Forest | XGBoost    | Transformer | EcoPerceiver |
> |------|--------------|---------------|------------|-------------|--------------|
> | CRO  | 4.4698       | 3.8319        | 3.2381     | 3.1873      | **2.8677**   |
> | CSH  | 2.1417       | 1.6411        | 1.5224     | 1.5613      | **1.4709**   |
> | CVM  | 6.0688       | 5.5726        | 5.5157     | 5.7824      | **5.2236**   |
> | DBF  | 5.3360       | 4.3661        | 4.0959     | 4.0451      | **3.8678**   |
> | DNF  | 4.2721       | 4.0485        | 4.0974     | 4.1143      | **3.6322**   |
> | EBF  | 6.1426       | 4.6365        | 4.6050     | 5.1420      | **4.3070**   |
> | ENF  | 3.7959       | 3.0497        | **2.8141** | 3.1755      | 2.8579       |
> | GRA  | 3.9406       | 3.3759        | 3.2487     | 3.6451      | **2.9059**   |
> | MF   | 4.9190       | 4.1002        | 3.8633     | 4.0511      | **3.7361**   |
> | OSH  | 2.7043       | 2.1384        | 1.8796     | 1.9609      | **1.7475**   |
> | SAV  | 2.3315       | 1.9455        | 1.6514     | 1.7497      | **1.3070**   |
> | SNO  | 1.7876       | 1.4126        | 1.4291     | 1.4873      | **1.1816**   |
> | WAT  | 5.3247       | 4.9366        | 3.1838     | **3.0019**  | 3.5802       |
> | WET  | 2.8352       | 2.5834        | **2.2073** | 2.2851      | 2.2830       |
> | WSA  | 2.8752       | 2.6903        | 2.5153     | 2.6952      | **2.4706**   |

---

> > ### Author Response · Authors · 2024-12-02
> > **Response to Reviewer yjf4**
> >
> > As we approach the end of the review period, we hope the additional information and experiments we've provided have addressed your concerns. If so, we kindly ask that you consider adjusting your score to reflect these updates. We hope that having this work accepted will go a long way toward sparking interest in this problem domain for the deep learning community, leading to better models and a better understanding of our biosphere.

---

### Meta-Review · Area_Chair_kywy · 2024-12-19

**Metareview:**

The paper presents an ML ready dataset for carbon flux modeling that includes multiple modalities (measure fluxes, satellite images, other predictors) making it a nice comprehensive dataset for use by the community. Further, they present a novel transformer model (based on the Perceiver architecture) to act as a baseline for this dataset and also show ablations regarding this model as well as comparisons to simple ML methods (linear regression, XGBoost, etc).

The main strength is the comprehensive dataset and all reviewers are in agreement regarding its utility for the community. The perceiver architecture is also well-motivated as a strong baseline to handle multiple modalities and missing data.
The main weakness is limited deep learning models used for evaluation of the dataset and a related limited analysis on the data itself (due to multiple modalities, missing data, data imbalance). While the second weakness is acknowledged partly by the authors in their limitation, an in-depth analysis on the effects of these could be useful - example, different DL models (conv-based, MLP-based, etc) may be more susceptible to this heterogeneity or be better choices and having them as part of the baselines may prove beneficial as well as an analysis as to why they are better/worse.

**Additional Comments On Reviewer Discussion:**

The reviewers mainly raised the limited evaluation of the models. All reviewers are agreed on the value of the dataset. Reviewers raised two main concerns: limited baselines/evaluations and limited ablations. For the limited evaluation, the authors added linear regression, RF, XGBoost and vanilla transformer. The first three are very simplistic models. More deep learning models as baselines would be useful, and relating the choice of these baselines to the heterogeneity of the dataset - multiple modalities, missing data, imbalanced data - would make the contributions stronger. For ablations, the authors have the ablations on components of the perceiver in the appendix.

ZVzD also raised the concern that the satellite images are of low resolution that could hinder conclusions but the authors's response on trade-off between more temporal snapshots and higher resolution to keep the dataset size manageable seems fair.

---

### Decision · Program_Chairs · 2025-01-22

Accept (Poster)